# Semiarid Lakes of Southwestern Siberia as Sentinels of On-Going Climate Change: Hydrochemistry, the Carbon Cycle, and Modern Carbonate Mineral Formation

Andrey Novoselov [1], Alexandr Konstantinov [2], Elizaveta Konstantinova [3], Yulia Simakova [4], Artem Lim [2], Alina Kurasova [2], Sergey Loiko [2] and Oleg S. Pokrovsky [2,5,*]

1. Earth Cryosphere Institute, Tyumen Scientific Centre SB RAS, Malygina 86, 625026 Tyumen, Russia; mr.andreygeo@mail.ru
2. BIO-GEO-CLIM Laboratory, National Research Tomsk State University, 36 Lenin Ave., 634050 Tomsk, Russia; konstantinov.alexandr72@gmail.com (A.K.); lim_artyom@mail.ru (A.L.); kurasovalina@gmail.com (A.K.); s.loyko@yandex.ru (S.L.)
3. Academy of Biology and Biotechnologies, Southern Federal University, 105/42 Bolshaya Sadovaya Str., 344006 Rostov-on-Don, Russia; konstantliza@gmail.com
4. Laboratory of Mineralogy, Institute of Geology of the Komi Science Center of the Ural Branch of the Russian Academy of Sciences, 54 Pervomayskaya Str., 167982 Syktyvkar, Russia; yssimakova@geo.komisc.ru
5. Geoscience and Environment Toulouse (GET), UMR 5563 CNRS University of Toulouse, 14 Avenue Edouard Belin, 31400 Toulouse, France
* Correspondence: oleg.pokrovsky@get.omp.eu

**Abstract:** Towards a better understanding of factors controlling carbon (C) exchange between inland waters and atmosphere, we addressed the inorganic carbon cycle in semiarid lakes of Central Eurasia, subjected to the strong impact of on-going climate change. As such, we assessed the hydrochemical variability and quantified its control on the formation of authigenic carbonate minerals, occurring within the upper layer of sediments in 43 semiarid lakes located in the southwest of Western Siberia (Central Eurasia). Based on measurements of pH, total dissolved solids (TDS), cationic and anionic composition, dissolved organic and inorganic C, as well as textural and mineralogical characterization of bottom sediments using X-ray diffraction and scanning electron microscopy, we demonstrate that lake water pH and TDS are primarily controlled by both the lithological and climatic context of the lake watershed. We have not revealed any direct relationships between lake morphology and water chemistry. The most common authigenic carbonates scavenging atmospheric $CO_2$ in the form of insoluble minerals in lake sediments were calcite, aragonite, Mg-calcite, dolomite and hydromagnesite. The calcite was the most common component, aragonite mainly appears in lakes with sediments enriched in gastropod shells or artemia cysts, while hydromagnesite was most common in lakes with high Mg/Ca molar ratios, as well as at high DIC concentrations. The relationships between mineral formation and water chemistry established in this study can be generalized to a wide suite of arid and semiarid lakes in order to characterize the current status of the inorganic C cycle and predict its possible modification under on-going climate warming such as a rise water temperature and a change in hydrological connectivity, primary productivity and nutrient regime.

**Keywords:** lakes; semiarid region; Western Siberia; inorganic C; $CO_2$; water chemistry; carbonates

## 1. Introduction

The water resources of Central Eurasia, located in the south of Western Siberia and Northern Kazakhstan, within a territory of more than 1,600,000 km² [1], are highly vulnerable in terms of their hydrological regime which is impacted by on-going climate warming. This region is characterized by a specific landscape mosaic, represented by a combination of narrow valleys of large rivers, their tributaries and isolated endorheic (seasonally dried) basins [2]. The hydrological structure of this vast territory, formed by a combination of

geomorphological and climatic factors, predetermined the wide distribution of lakes and ponds—most of them are located within the drainless watersheds [3,4]. The total number of identified lakes (area > 1 km$^2$) in the south of Western Siberia is estimated as more then 20,000. The overall limnicity of the territory ranges between 1 and 10% [5], which is unusually high for the relatively arid region of the world. Small (<1 km$^2$) water bodies are strongly predominant in overall lake number. Most of the small lakes are saline, with overall mineralization higher than 1 g L$^{-1}$ [6,7]. These lakes are of special interest, because these systems are very sensible to natural and human-induced changes in the climate and transformation of the landscapes. Moreover, saline and hypersaline aquatic environments of drainless basins are potentially important for biodiversity and ecological studies [8]. Further, small water bodies play a disproportionly important role in the terrestrial geochemical cycles of C and related elements [9,10]. A number of recent studies have shown that the role of small lakes and ponds of the arid and semiarid zones in controlling the continental fluxes of elements, notably carbon, is often underestimated [11].

The carbonate system of the lake, including the water column and sediments, is one of the most important factors that predetermines the potential of these water bodies for carbon uptake and storage [12]. The rate of insoluble carbonate mineral formation in lakes is related to a number of parameters including geological and geomorphological context, the hydrochemical composition of waters, hydrological regime, biological activity, as well as the erosional processes within the watersheds [13]. Studies performed for large endorheic lake systems in different arid and semiarid regions of the world have shown that they are characterized by a high degree of intergroup variability in terms of sedimentation processes and authigenic mineral formation [7,14–25]. These studies support the idea that the analyses of the role of endorheic lakes in the formation of regional terrestrial geochemical fluxes primarily requires an understanding of their morphometric diversity taking into account hydrochemical parameters and intensity of C sink in sediments.

The forest-steppe and steppe lakes of Western Siberia are common within a vast territory extending over approximately two thousand kilometers from west to east and are confined to several large geomorphological regions: the Trans-Urals, the Ishim plain, the Barabinsk lowland and the Kulunda plain [7]. The hydrochemical features and sedimentation processes in water bodies of the Trans-Urals and the Ishim plain are much better studied compared to the Barabinsk lowland and the Kulunda plain, though they occupy a significant part of the semiarid part of Western Siberia. Only a few studies were devoted to assessing the composition of waters and bottom sediments of these geomorphological regions with a focus on water bodies, where the formation of authigenic Mg and Mg-Ca carbonates occurs [26–28].

The present study is devoted to assessing the hydrochemical variability of small endorheic water bodies of the Trans-Urals and the Ishim plain geomorphological regions using an example of five groups of lakes, illustrating the gradual physicogeographical transition from Trans-Urals to the inner part of the West Siberian plain. Given that carbon dioxide (CO$_2$) exchange with the atmosphere and C burial in the form of organic and inorganic carbon are strongly coupled via aquatic photosynthesis, respiration and authigenic carbonate mineral dissolution and precipitation, we focused on assessing the major hydrochemical and mineralogical features of the contrasting and yet representative lakes of the region. Our main objectives were to: (*i*) evaluate the spatial, morphological and hydrochemical variability of the studied lakes; (*ii*) analyze the factors affecting the differences in the chemical composition of waters between different groups of lakes; (*iii*) characterize the typical associations of authigenic carbonates as well as to discuss the possible factors affecting their formation and diversity in sediments. Collectively, these results should enable a better understanding the role of small lakes in the C cycle between the land and the inland water bodies in the context of their strong vulnerability to on-going climate warming in central Eurasia.

## 2. Materials and Methods

### 2.1. Geological and Geographical Setting

The studied lakes are located in the forest zone of southwestern Siberia within the territories of the Chelyabinsk, Kurgan and Tyumen regions of Russia (Figure 1). The water bodies are grouped into five key sites that form a transect from the Ural Mountains to the Ishim Plain. In terms of the area of the water surface, 2.3% of the studied lakes are medium water bodies (1–10 km²), 4.6% are small water bodies (0.1–1 km²), and the remaining 93.1% are very small water bodies (<0.1 km²). All studied lakes are rather shallow, with an average depth not exceeding 3 m (Table A1). The duration of the ice cover on studied lakes ranges from 150 to 180 days, depending on the mineralization, size, depth and exact localization within the study territory [26].

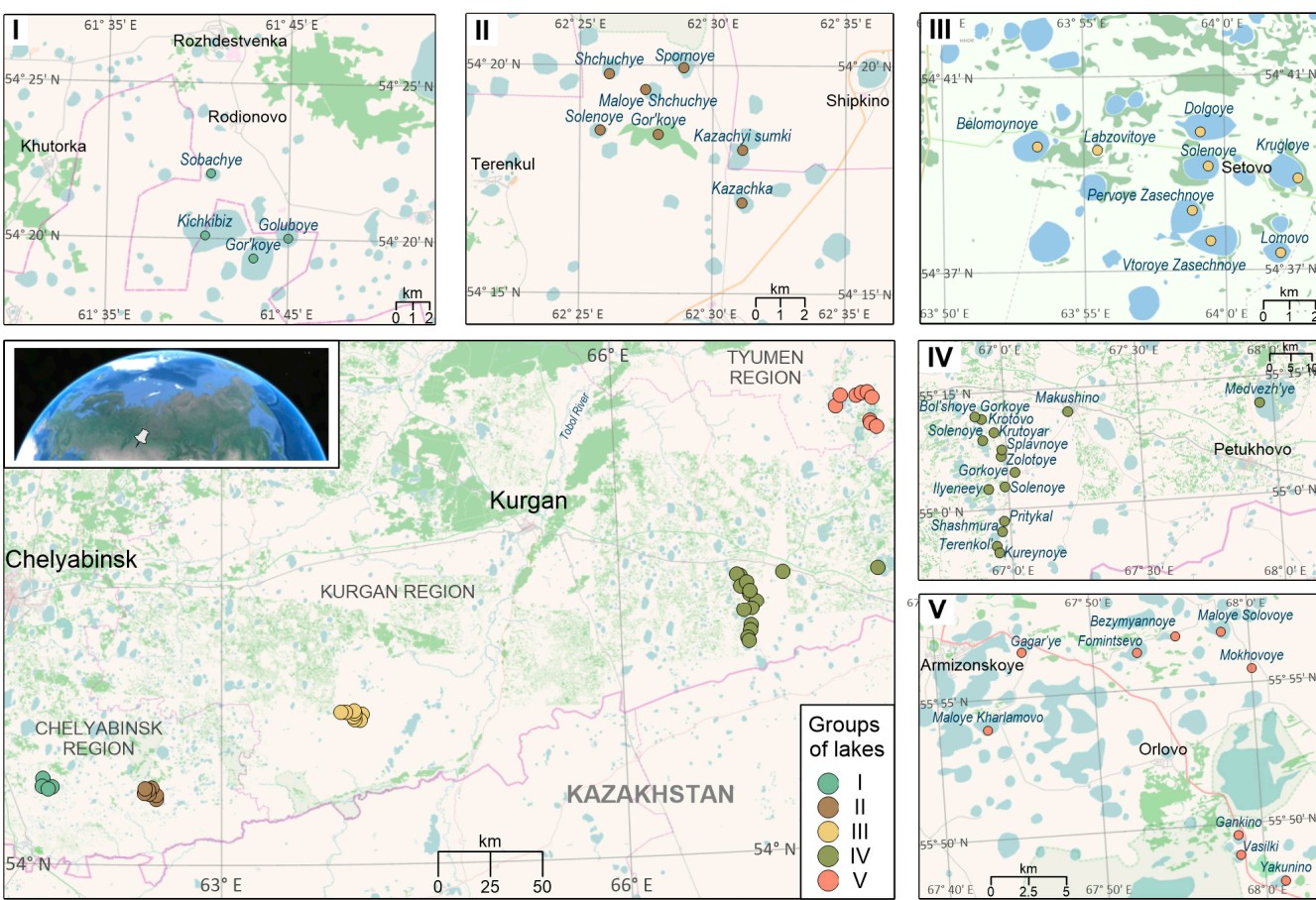

**Figure 1.** The location of the studied lakes in the southwest of Western Siberia (base maps from www.openstreetmap.org, accessed on 20 August 2023).

The territory belongs to the southwestern edge of the West Siberian Plain, which is a relatively flat area, with <50 m elevation differences and a general slope from south to north and from west to east. The western and the central parts of the territory correspond to the Trans-Ural plain and the eastern to the Ishim plain. The absolute heights decrease form 170–190 m a.s.l. within Chelyabinsk region to 140–150 m within the Kurgan region and 120–130 m within the Tyumen region. The relief of both Transuralian and Ishim plains is weakly dissected, represented by alternating flat watersheds and depressions. The similarity of the geomorphological features of this territory predetermined the wide distribution of drainless basins with lake depression. Suffusion lake depressions are the most common within the discussed area [2].

The river network is poorly developed. Except for the Tobol River valley and its large tributaries, most of the territory belongs to the endorheic basin [2]. The territory under

consideration belongs to the subzone of the forest-steppe. The climate is continental with an annual temperature 2.7–2.9 °C for the Trans-Ural plain and 1.7 °C for the Ishim plain. Annual precipitation varies from 436 mm in the western part of the study area region to 383 mm in the central part and 393 in the eastern [29]. The landscapes of the study territory are significantly transformed by anthropogenic activities, mainly, agriculture. The watersheds are covered with birch groves and plowed fields [30]. The soil cover is represented by Chernozems on the watersheds and Planosols in depressions, which is typical for the West Siberian forest-steppe ecotone [31]. The near-shore zones of saline lakes are often occupied by Solonetz and Solonchak soils [27].

The underlying deposits of the whole studied transect from Trans-Ural plain to Ishim plain are rather similar in terms of genesis and are represented by Quaternary loams of the Uisko-Uboganskaya suite (LIIuu, up to 15 m) and the carbonate loess-like loams of the Zyryansk horizon (LIIIzr). Quaternary deposits are underlain by sandy and loamy strata of the Miocene, Oligocene, and Eocene [32]. The thickness of Quaternary deposits gradually increases with distance from Urals. The aquifers of the study territories are represented by mainly Cretaceous and Eocene-Paleocene marine strata located at a depth of more than 100 m. The underground waters of these horizons are characterized by low mineralization (<1–2 g L$^{-1}$), chloride-hydrocarbonate composition with a low content of magnesium ions [26].

### 2.2. Field Work and Sampling

We have sampled 43 water bodies in the Chelyabinsk, Kurgan and Tyumen regions of Russia, representing five groups of lakes that illustrate the gradual transition from the western periphery of the West Siberian plain to its inner part. In total, 43 samples of water and 36 samples of sediments collected during fieldwork periods of 2018–2019 were analyzed in this study (Figure 1, Tables A1 and A2). Sampling field campaigns were mainly conducted in summer and autumn periods corresponding to summer baseflow of the region, which is the main period of authigenic carbonate mineral formation [27,28]. One of the lakes (Zolotoye) was studied in early spring. The two sampled years were within the mean multi-annual air temperature and precipitation. Indeed, mean annual air temperatures of 2018 and 2019 years were 2.1 and 3.3 °C, respectively, while for the historic period of observations the MAAT is 2.9 °C. Mean annual precipitation values of 2018 and 2019 were 343 and 406 mm, respectively, which is comparable with mean mean-multi-annual value of 391 mm. These values have to be considered with caution, given that the study area is rather large and extended; as such, the data obtained at the weather station of the largest regional center (Kurgan) cannot be directly extrapolated to remote parts of the studied territory.

Water samples for chemical analysis were collected from the 40 to 50 cm depth. Temperature (T), pH and total dissolved solids (TDS) were measured in situ using a Multi 3420 portable multiparameter meter (WTW, Xylem Analytics, Germany). The water was filtered on-site through a 0.45 μm membrane filter (MILLEX Syringe Filter unit, Merck, Germany). The first 20–50 mL of filtrate was discarded and the subsequent filtrate was collected into pre-washed 250 mL polypropylene bottles. All the chemical vessels were preliminarily washed and sterilized to prevent contamination. Filtered water samples were taken in two bottles, one was acidified by ultrapure concentrated HNO$_3$ for the analysis of cations, and the second was not acidified and used for analyses of dissolved organic (DOC) and inorganic (DIC) carbon and anions [33]. Before analyses, filtered samples were stored in a refrigerator at 4–6 °C.

Sediment samples were collected using an Ekman Bottom Sampler. Upper 10 cm of the sediment core were used for laboratory analysis. The sediment samples were air-dried, mixed, ground and passed through a 1 mm sieve for particle size analysis [34].

### 2.3. Hydrochemical Analysis

Water samples were diluted based on the salinity prior to water chemistry analysis. Concentrations of the major anions ($Cl^-$ and $SO_4^{2-}$) were analyzed by ion chromatography (Dionex 2000i, Thermo Fisher Scientific, Waltham, MA, USA) with an uncertainty of 2%. The dissolved organic carbon (DOC) and dissolved inorganic carbon (DIC) were determined using a TOC-Vscn analyzer (Shimadzu, Kyoto, Japan) with an uncertainty of 3% and a detection limit of 0.1 mg/L. The samples were diluted by a factor of 3 to 10 prior to analyses. Concentrations of the major cations ($Mg^{2+}$, $Ca^{2+}$, $Na^+$ and $K^+$) were measured by quadrupole Agilent 7500ce ICP-MS system (Agilent Technologies, Santa Clara, CA, USA) with an uncertainty of $\pm 5\%$. Indium and rhenium were used as internal standards. The international geostandard SLRS-5 (Riverine Water Reference Material for Trace Metals, certified by the National Research Council of Canada) was used to check the validity and reproducibility of each analysis.

### 2.4. Mineralogical and Textural Analysis

The particle size distribution of sediment samples was analyzed using a laser diffraction analyzer (LS 13 320 Beckman Coulter, Brea, CA, USA), with preliminary sample dispersion in sodium pyrophosphate. Particle size fractions were defined as clay (<0.002 mm), silt (0.002–0.05 mm), and sand (0.05–2.0 mm) according to the USDA particle size classification [35].

The dried samples of sediments were used for mineralogical studies via scanning electron microscopy–energy dispersive spectroscopy (SEM-EDS). The SEM-EDS analysis was performed using a TM3000 scanning electron microscope (Hitachi, Tokyo, Japan) with a Quantax 70 EDS attachment (Bruker, Billerica, MA, USA) at X100–5000 magnification and a JSM-6390LV scanning electron microscope (Jeol, Tokyo, Japan) with an INCA Energy 450 X-Max80 EDS attachment (Oxford Instruments, Abingdon, UK). The SEM observation were made under high vacuum (HV-mode), mainly in the elemental composition mode (BSE, registration of back scattered electrons). While performing the EDS analysis, the voltage was 15 and 20 kV for the first and second devices, respectively.

The semi-quantitative mineralogical composition of the collected sediments was examined by a powder X-ray diffraction (XRD) technique. The analysis was performed with a Shimadzu XRD-6000 diffractometer (Shimadzu, Kyoto, Japan), using Co-K$\alpha$-radiation (Ni filter, 30 kV, 20 mA) at a 2θ range from $10°$ to $60°$ and a scan speed of $0.03 \text{ s}^{-1}$. Semiquantitative phase analysis was performed using diffraction patterns of non-oriented samples in the PROFEX program. Corundum standards were always used to constrain the calibration. Analytical uncertainty of the quantification was within 1 wt.%. However, in case of very low content of a given mineral phase in the bulk sample (i.e., 2 to 3%), the uncertainty on its quantification could reach $\pm 100\%$.

### 2.5. Statistical Treatment and Data Visualization

Calculation of saturation indices of aqueous solution with respect to various carbonate minerals was performed using the Visual MINTEQ ver. 3.1 [36]. The parameters including pH, temperature, and the concentrations of individual ions obtained in situ were used as input parameters.

The software package STATISTICA 12 (StatSoft, Tulsa, OK, USA) was used to analyze data. Basic descriptive statistics included the mean, median, minimum, maximum, standard deviation (SD), and coefficient of variation (CV) for all hydrochemical parameters.

The relationship between the morphometric parameters of the lakes and the hydrochemical properties of the waters was assessed using correlation analysis with the Spearman rank correlation coefficient (significance level $p < 0.05$). The Kruskal–Wallis one-way analysis of variance (ANOVA) by ranks and the median test followed by multiple comparisons of mean ranks for all groups were applied to evaluate differences in hydrochemical properties of the five groups of lakes. Regression analyses were also used to determine the relationship between the main hydrochemical parameters. Data visualization

was performed using Grapher 17 (Golden Software, Golden, CO, USA) and STATISTICA 12 (StatSoft, USA) software.

## 3. Results

### 3.1. Water Chemistry

The data on the water chemistry of the studied lakes are listed in Table A2 of the Appendix A. In the summer–autumn season, the water temperature averaged 19.5 °C; the maximum value of 24.9 °C was recorded in September in Lake Solenoye (group III). Water chemistry varied significantly among all studied water bodies (Table 1), which indicates the high degree of heterogeneity, despite of rather similar geological and lithological context. Only in 35% of the studied lakes, the pH values were in the range of 6.5–8.5, which corresponds to a neutral and weakly alkaline reaction. The rest of lakes were characterized by a strongly alkaline solution (8.9 < pH < 10.4). In general, the studied water bodies were characterized by high mineralization; the TDS averaged 26.9 g $L^{-1}$. Fresh water type (TDS < 1 g $L^{-1}$) referred to 32.6% of the studied lakes, slightly saline water type accounted for to 4.6% (1–3 g $L^{-1}$), moderately saline water type amounted to 16.3% (3–10 g $L^{-1}$), highly saline water type accounted for 16.3% (10–35 g $L^{-1}$), transitional to brines type constituted 11.6% (35–50 g $L^{-1}$) and brine type amounted to 18.6% (>50 g $L^{-1}$). The median concentrations of dissolved major constituents followed the order Na > Mg > K > Ca and Cl > $SO_4$ > DIC > DOC.

**Table 1.** Descriptive statistics for main water physicochemical parameters of the semiarid lakes in southwestern Siberia (n = 43).

| Parameter | Unit | Mean | Med | Min | Max | SD | CV, % | SE |
|---|---|---|---|---|---|---|---|---|
| pH | – | 8.7 | 8.9 | 6.9 | 10.4 | 0.8 | 8.8 | 0.1 |
| TDS | g $L^{-1}$ | 26.9 | 9.8 | 0.1 | 160 | 37.1 | 138 | 5.7 |
| $Mg^{2+}$ | | 1260 | 233 | 6 | 11,200 | 2260 | 179 | 344 |
| $Ca^{2+}$ | | 103 | 46.9 | 7.4 | 669 | 147 | 142 | 22 |
| $Na^+$ | | 7100 | 2050 | 20.1 | 55,600 | 11,500 | 162 | 1760 |
| $K^+$ | mg $L^{-1}$ | 84.9 | 51 | 7.5 | 439 | 85.9 | 101 | 13 |
| $Cl^-$ | | 9510 | 2270 | 8.6 | 83,100 | 15,800 | 166 | 2400 |
| $SO_4^{2-}$ | | 4010 | 330 | 0.3 | 63,900 | 10,300 | 258 | 1600 |
| DOC | | 80.3 | 47.1 | 11.9 | 1230 | 182 | 227 | 28 |
| DIC | | 144 | 101 | 20 | 494 | 116 | 81 | 18 |

The relative ionic composition of water samples representing all studied lakes is plotted on a ternary Piper diagram (Figure 2). According to the ratio of cations, most of the lakes are of the sodium-potassium type. For some water bodies, mainly of group V, the dominant type of cationic composition is not distinguished. According to the anionic composition, most of the lakes of groups I to IV are of the chloride type, while among the lakes of group V, bicarbonate waters predominated. Thus, most of the studied lakes were characterized by sodium chloride waters, some lakes of the V group belonged to the magnesium bicarbonate type.

It should be noted that water chemistry varied widely within the same geographical group, which indicates that the climate and geological context do not have a direct effect for the differentiation of hydrochemical parameters. However, according to the Kruskal Wallis test and the median test, there were significant differences between the mean values of TDS, major ion content, DOC and DIC ($p < 0.005$) in waters belonging to the certain lake group (Table A5). Thus, group III lakes are characterized by maximum median TDS values (42 g $L^{-1}$), corresponding to increased water salinity, $Na^+$ (8900 mg $L^{-1}$), $Cl^-$ (12,000 mg $L^{-1}$), DOC (91.7 mg $L^{-1}$), and DIC (309 mg $L^{-1}$) (Figure 3). In lakes of group I, elevated median values of $Mg^{2+}$ (1400 mg $L^{-1}$), $K^+$ (173 mg $L^{-1}$), and $SO_4^{2-}$ (3495 mg $L^{-1}$) are observed. The minimum median values of all studied physicochemical parameters, except for DIC, were characteristic of lakes of the V group.

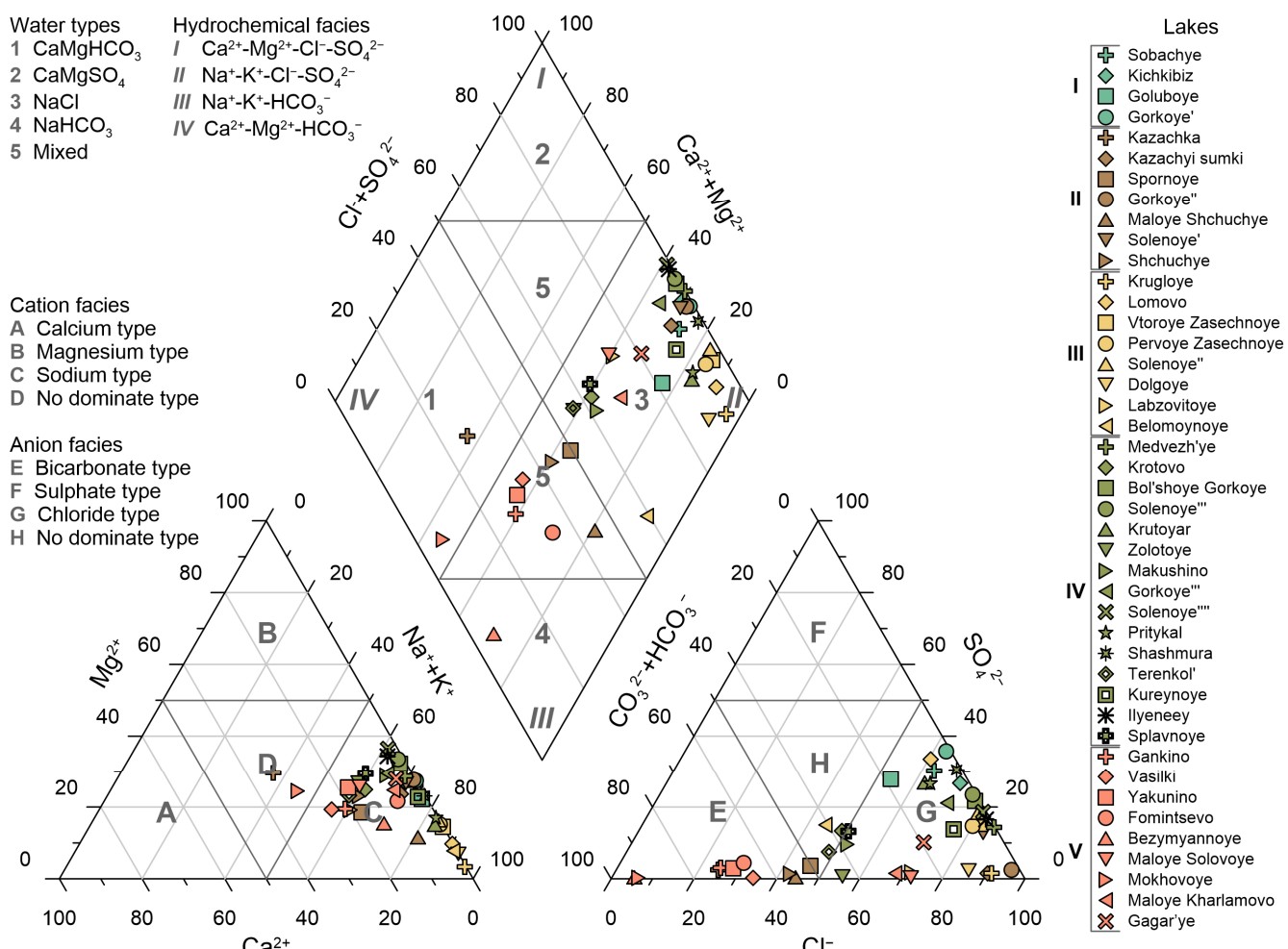

**Figure 2.** Piper diagram of the water chemical data from semiarid lakes in the southwest of Western Siberia. Groups of lakes denoted by color and roman numerals. Note that the DIC is represented by the sum of $HCO_3^-$ and $CO_3^{2-}$ ions.

The relationship between the morphometric characteristics of lakes and the physico-chemical parameters of the water column was assessed based on the analysis of Spearman correlation coefficients (Table A6). Statistically significant ($p < 0.05$) weak and moderate positive correlations ($0.36 < r < 0.52$) were revealed between the lake area and TDS and the concentration of main cations and anions. These relationships can be explained by different volumes of water evaporating from the surface of lakes having different size, which leads to uneven levels of seasonal concentration of solutes. Moreover, the different area of the water bodies controlled the potential role of groundwater and its connectivity to the water column of the lake. The contribution of precipitation and surface runoff depended on the watershed area. There were also weak but positive correlations between catchment area and major ion content ($0.4 < r < 0.5$, $p < 0.05$). At the same time, the average depth of water bodies did not control the water chemistry ($r < 0.3$; $p > 0.05$). A moderate inverse relationship was noted between pH values and Ca concentration in the lake water ($r = -0.53$, $p < 0.05$), which can be explained by the fact that dissolved $Ca^{2+}$ ions are able to remove the excess of dissolved carbon dioxide, lowering the acidity of the waters [37]. The TDS values demonstrated strong link ($r > 0.9$, $p < 0.05$) with the concentration of $Cl^-$, $Na^+$, $Mg^{2+}$, $SO_4^{2-}$, DOC and strong correlation ($r = 0.82$, $p < 0.05$) with the concentration of $K^+$. The maximum r value was observed between $Na^+$ and $Cl^-$ (0.98). It was previously shown that chlorides dominated the mineralization for the studied water bodies. Moderate positive correlations were noted between TDS with $Ca^{2+}$ and DIC ($0.5 < r < 0.7$). Major cations and anions, DOC

and DIC are mutually correlated with each other at the $p < 0.05$ level. Additional regression analysis was carried out between the most important components of the carbon cycle in the lakes. The DOC was moderately correlated with $Mg^{2+}$ ($r^2 = 0.59$, $p < 0.00001$). However, there was no linear relationship of DOC with $Ca^{2+}$ ($r^2 = 0.002$, $p = 0.785$). The DIC was related to none of these inorganic components ($r^2 < 0.12$, $p > 0.02$), which likely stems from complex salt composition and the history of the formation of the studied lakes.

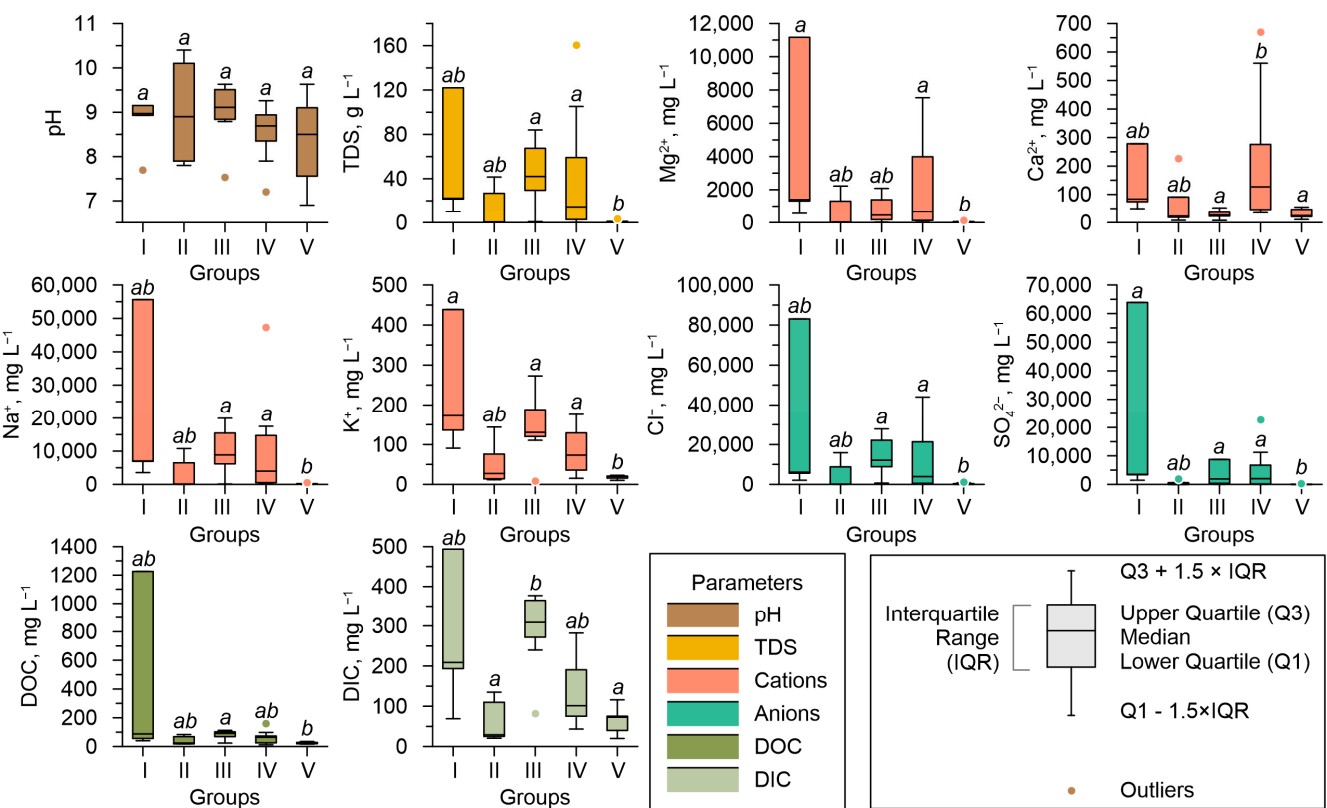

**Figure 3.** Variation in hydrochemical properties between groups of the semiarid lakes of the southwest of Western Siberia. Different letters indicate significant differences ($p < 0.05$) resulting from the multiple comparisons of $p$-values.

## 3.2. Water Chemistry as a Factor of Mineral Formation

The values of Mg/Ca molar ratio fluctuated over a wide range, from 0.8 to 198 for all of the studied water bodies. In 35% of the lakes, the Mg/Ca molar ratio was below 2. These lakes also demonstrated lower TDS values (<1 g L$^{-1}$). For 35% of the studied water bodies, the Mg/Ca molar ratio was above 10. These lakes were also characterized by elevated TDS values, though no pronounced relationships between this Mg/Ca molar ratio and TDS were observed (Figure 4). A regression analysis demonstrated that there are significant ($p < 0.05$) linear relationships between Mg/Ca molar ratio, TDS and DOC/DIC. At the same time, the identified relationships between mentioned parameters and Mg/Ca molar ratio were rather weak ($R^2 < 0.5$).

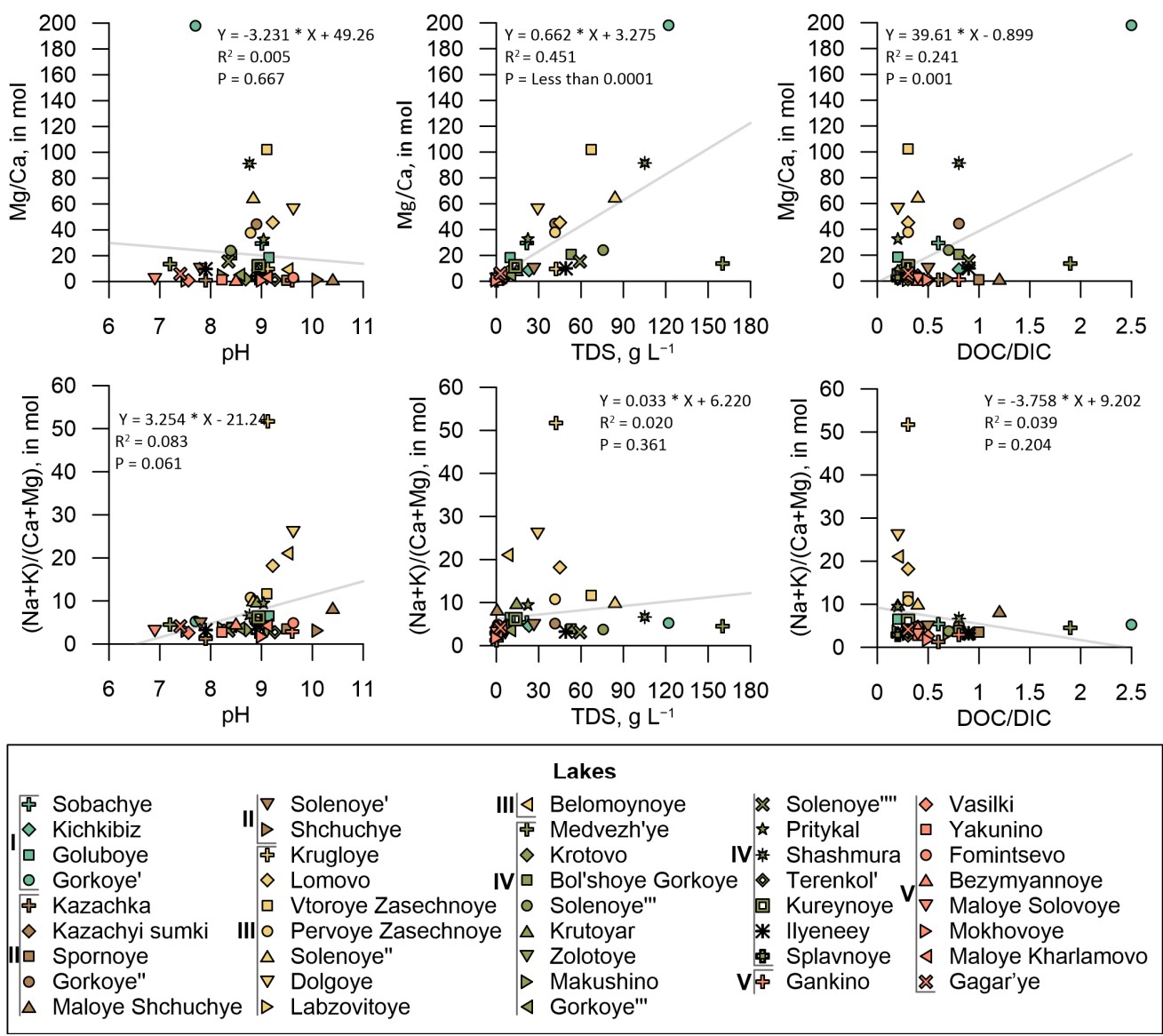

**Figure 4.** Mg/Ca and (Na+K)/(Ca+Mg) molar ratios plotted against pH, TDS, and DOC/DIC in water of the semiarid lakes of the southwest of Western Siberia. Groups of lakes denoted by color and roman numerals.

The (Na+K)/(Ca+Mg) molar ratio varied from 1.2 to 52 among all studied water bodies. At the same time, no statistically significant linear relationships between the values (Na+K)/(Ca+Mg) ratio and other hydrochemical parameters were observed (Figure 4).

Thermodynamic modelling of water–rock equilibrium performed using the Visual MINTEQ allowed to identify possible secondary carbonate minerals in the studied lakes of different types. Their water saturation indices (SI) are listed in Table A3 of the Appnedix. The saturation indices (SI) of minerals systematically changed depending on the pH and TDS values (Figure 5). According to the Visual MINTEQ results, as pH increases, carbonates have a possibility to precipitate in the following order: dolomite ($CaMg(CO_3)_2$, ordered and disordered), aragonite ($CaCO_3$), calcite ($CaCO_3$), vaterite ($CaCO_3$) at pH > 7.9; magnesite ($MgCO_3$), huntite ($Mg_3Ca(CO_3)_4$ at pH > 8.2; monohydrocalcite ($CaCO_3xH_2O$) at pH > 8.9. Artinite ($Mg_2CO_3(OH)_2 \cdot 3H_2O$) and hydromagnesite ($Mg_5(CO_3)_4(OH)_2 \cdot 4H_2O$) could precipitate at pH 8.7–9.2. Regardless of the salinity in the solution, precipitation of dolomite (ordered and disordered), aragonite, calcite, vaterite is possible for all studied aquatic environments. At TDS > 1 g $L^{-1}$, the solutions become oversaturated, and magnesite and

huntite precipitation is also possible. Monohydrocalcite, artinite and hydromagnesite can precipitate in highly lakes with saline waters.

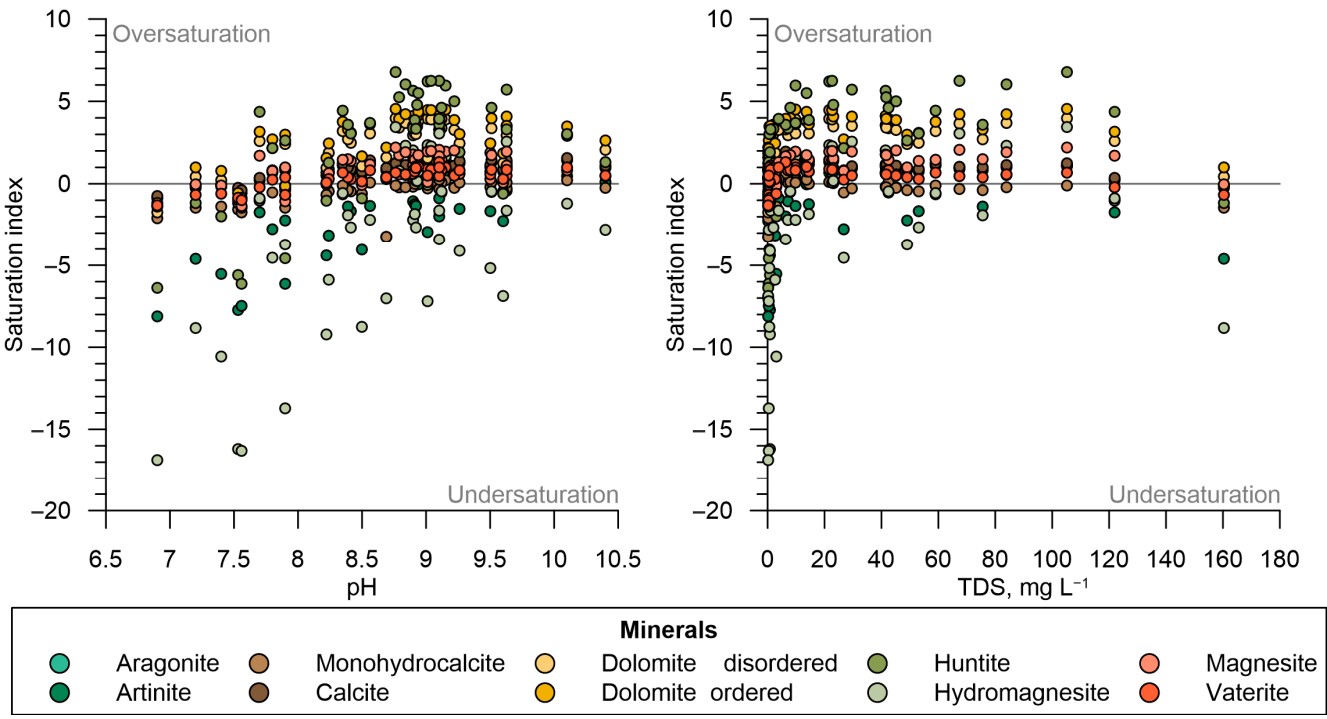

**Figure 5.** Saturation indices of carbonate minerals plotted versus pH and TDS values.

### 3.3. Lithology and the Bulk Mineralogical Composition of Sediments

In most cases, the upper layer of sediments was represented by a dark-grey organic-rich material, varying between different lakes depending on the proportion of mineral matter. According to the textural differentiation, the upper layer of sediments exhibited distinct granulometric composition (Figure 6). Sediments from lakes of I–III and V groups were characterized by a significant proportion of sand fraction, while the group IV lakes sediments referred to loams with a sizable proportion of silt fraction.

Based on the results of XRD and SEM-EDS analytical studies, we have identified terrigenous and authigenic minerals constituting the sediments of forest-steppe lakes of the southwest of Western Siberia (Table A4 of the Appendix A). For the majority of the studied aquatic environments, quartz and, to a lesser extent, feldspars prevailed among the terrigenous constituent with a small admixture of muscovite, biotite, accessory minerals and other aluminosilicates.

For most of the sediment samples, minerals from the group of clay aluminosilicates such as smectite-montmorillonite, illite and chlorite were identified. Clay minerals were identified in non-oriented samples by non-basal reflections d/n~4.45, 2.55 Å. Illite basal reflections~10.1–10.4 Å, smectite d001~13.8–14.5 Å, chlorite-d001~14.2–15 Å, mixed-layer phases were also present. According to the results of microscopic studies, there were clear signs of the authigenic formation of chlorite-hydromicaceous material in cavities and caverns of the substrate mass. This suggested the authigenic formation of clay minerals in lacustrine sediments, that can be intensified during the recent period as a result of enhanced erosion caused by agricultural activities within the watersheds as known in other desert and semi-desert settings [38].

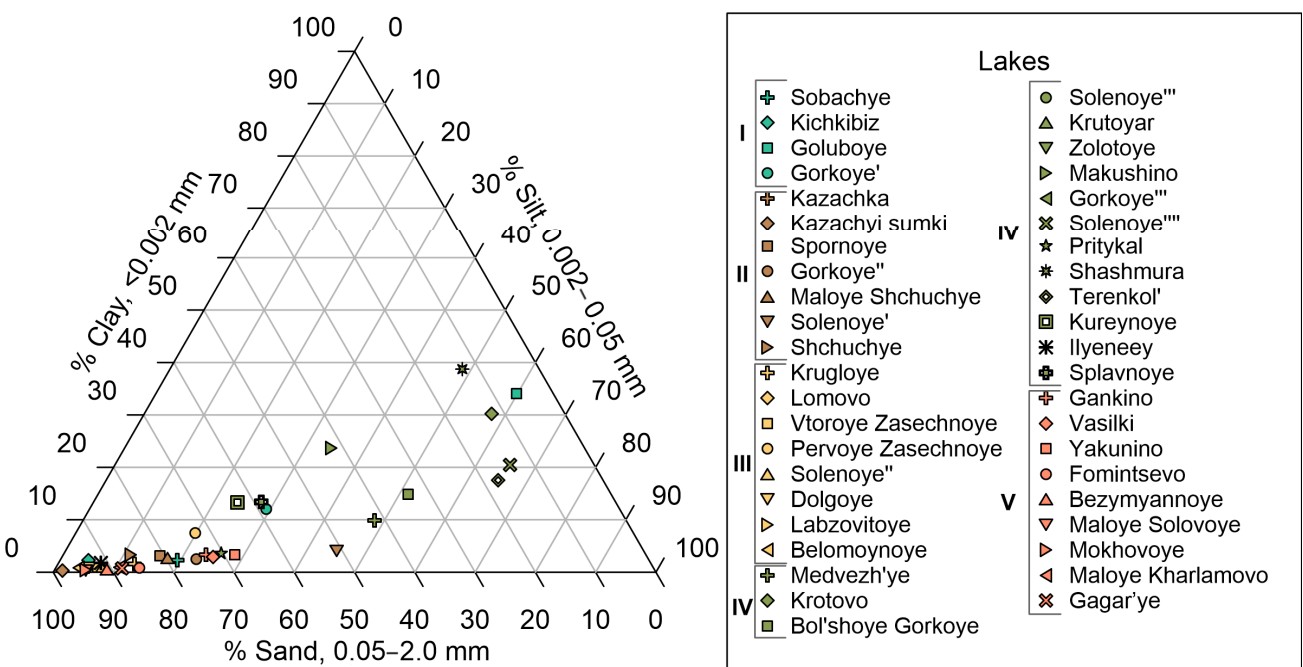

**Figure 6.** Sand-silt-clay values for 36 sediment samples plotted on the USDA textural triangle.

Single aggregates of oxides and hydroxides of iron are found in sediments in insignificant amounts, mainly presented in the form of films on the surface of terrigenous grains and aggregates of hydromicaceous material. It is possible that the authigenic formation of iron oxides within the mass of the studied sediments was associated with post-sedimentary replacement of organic matter in the form of finely dispersed detritus and organic films [39,40].

The group of sulfate minerals was mainly represented by gypsum, whose formation could be linked to intensive fluctuations of water level during the summer period. As a rule, sulfates within the studied bottom sediments were represented by both single elongated columnar and flat microcrystals and aggregates of various configurations, most often in the form of rounded acicular gypsum microconcretions. For some lakes with highly mineralized brines, aqueous magnesium and sodium sulfates were identified, represented mainly by bloedite ($Na_2SO_4 \times MgSO_4 \times 4H_2O$) in lakes of I group, as well as loeweite ($Na_4Mg_2(SO_4) \times 4.5H_2O$), also reported in other lakes of the region [41] and kieserite ($MgSO_4 \times H_2O$) in lakes of III group. Among the authigenic minerals of the halide group, we identified only halite (NaCl).

### 3.4. Abundence and Diversity of Carbonate Minerals in Bottom Sediments of the Studied Lakes

Among the variety of authigenic minerals in sediments of the studied water bodies, carbonates hold a specific place in the context of our research. This class of minerals is found in almost all water bodies (Table A4 of the Appendix A). The most common carbonates include calcite ($CaCO_3$), Mg-calcite ($Ca,Mg)CO_3$), aragonite ($CaCO_3$), Ca-dolomite ($Ca,Mg(CO_3)_2$) and hydromagnesite ($Mg_5(CO_3)_4(OH)_2 \times 4H_2O$); and in rare cases, siderite ($FeCO_3$) and rhodochrosite ($MnCO_3$). The composition of carbonates is quite dynamic, diverse, and is largely determined by a combination of features of specific aquatic environment.

The XRD patterns of most widespread carbonates are present in Figure 7. Calcite is the most common carbonate mineral which is present in sediments of all studied groups of lakes. The highest abundance of this mineral was observed for IV and V groups of lakes. According to the results of SEM-EDS analysis, calcite often forms pelitomorphic nodules within the clayey mass or it appears as intergrowths of well-formed rhombic isometric microcrystals. Aragonite is a rather common component of bottom sediments enriched in

gastropod shells, as well as artemia cysts (most of the brine-type and highly mineralized lakes of I and III group and several eutrophic lakes of the II, IV and V groups).

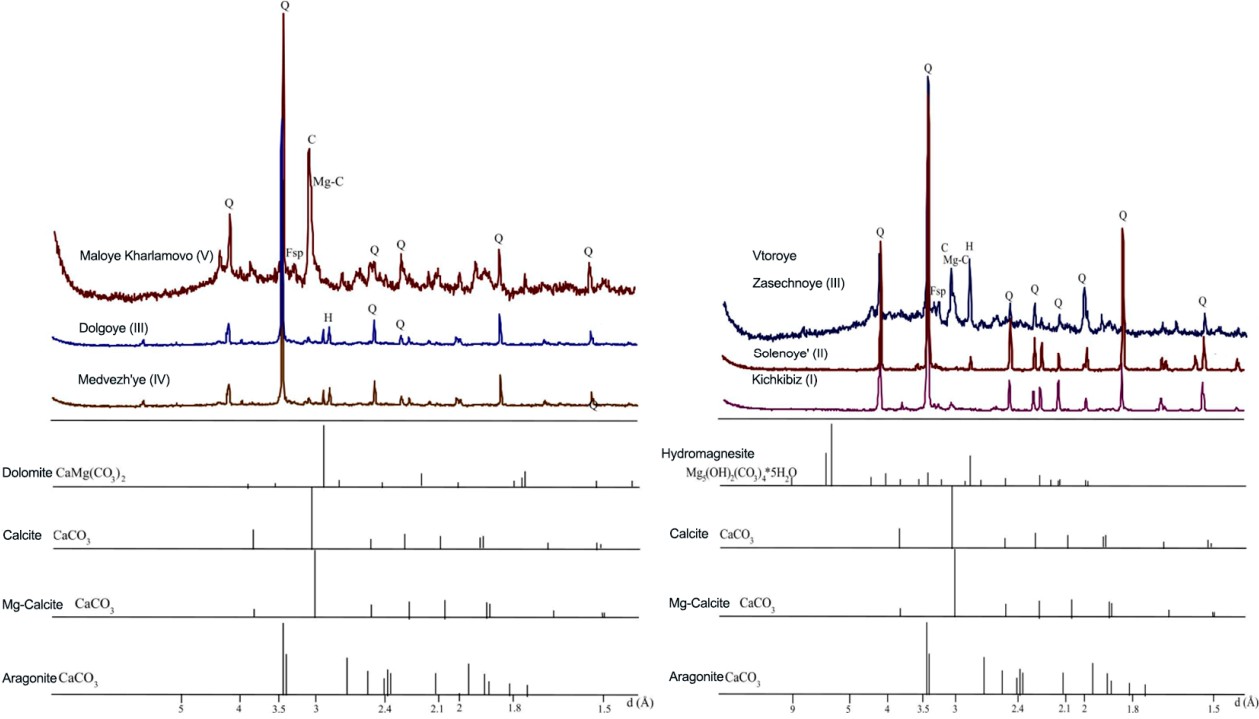

**Figure 7.** XRD patterns of most common carbonates recorded from lakes representing various groups.

For a large number of samples, the broadening of reflections suggests the presence of several calcite phases that differ in Mg content. According to the results of the XRD analysis, there was a continuous isomorphic miscibility for calcite-magnesite series with a corresponding change in the interplanar distances of minerals; therefore, the diffraction patterns of Mg-calcite differ from those of calcite. Mg-calcite is observed in variable amounts in sediments of all lakes. Basically, this mineral formed aggregates with hydromagnesite or proto-dolomite, most often it formed large communities of single elongated split sheaf-like microcrystals that develop within the EPS films, rarely forming rather large poorly cemented homogeneous nodules.

Dolomite appeared in sediments of III and IV group lakes in small amounts. Given intrinsic limitations on quantification of mineral proportion by the XRD analysis ($\pm100\%$ for 2–3% of total fraction), we can only discuss the abundance of this component. At the same time, the SEM-EDS microanalyses have shown that this mineral appears as small inhomogeneous microcrystalline concretions, mainly in a mixture with high-magnesian calcite (Figure 8). In several cases there were dolomite microcrystals formed during the mineralization of EPS films in the form of single scattered cubic-like microcrystals. More rarely, dolomite appeared as large aggregates, consisting of co-directional columnar crystals with a high content of magnesium (17.6% mass fraction of Mg and 2.5% for Ca). In such cases, non-mineralized carbon-rich biogenic microfilms were observed on the surface of the aggregate and between individual columnar crystals, which is a commonly described pathway of Mg-rich carbonates precipitation in aquatic environments [42,43].

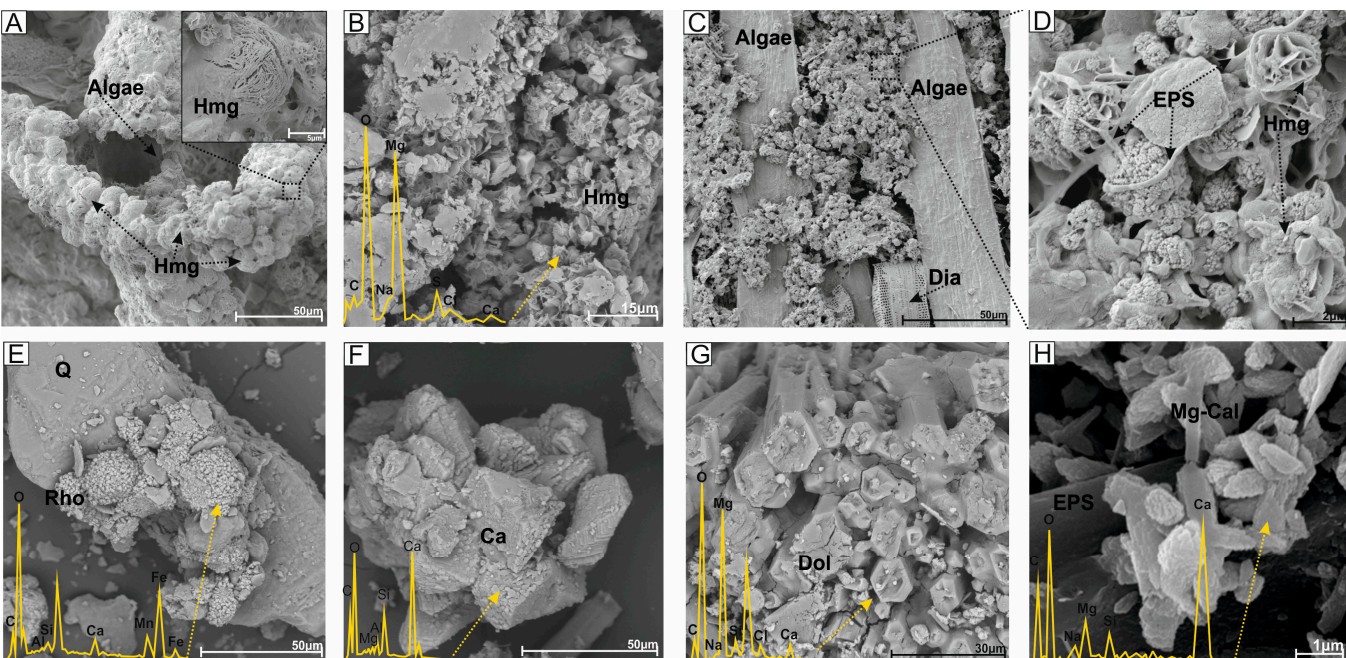

**Figure 8.** SEM images illustrating authigenic carbonates: (**A**) Mineralized algal filaments covered with a shell of spherical microaggregates of hydromagnesite (Lake Vtoroye Zasechnoye). (**B**) massive hydromagnesite aggregates (Lake Dolgoye). (**C**,**D**) Microaggregates of hydromagnesite formed on the surface of living algal filaments along EPS films (Lake Solenoye"). (**E**) Spherical mi-croaggregates of high-iron rhodochrosite on the surface of terrigenous quartz grains (Lake Kich-kibiz). (**F**) Intergrowths of microcrystals of authigenic calcite (Lake Kichkibiz). (**G**) Intergrowths of columnar (columnar) microcrystals of high magnesian dolomite (Lake Goluboye). (**H**) Split sheaf-shaped microcrystals of dolomite, formed on EPS films (Lake Medvezh'ye).

Hydromagnesite occurred in sediments of a number of water bodies from I, II and III groups. It was rather common for the lakes of the third group, where it was previously reported as characteristic mineral of near-shore facies and microbialites [26,41]. According to the results of SEM-EDS analysis, this mineral appeared as continuous zones of plate-like radially divergent crystals, associated in dumbbell or star-like clusters, that develop over extracellular polysaccharides (EPS) films. This form is rather common for this mineral, which often occurs in contemporary aquatic settings [44–47].

## 4. Discussion

### 4.1. Hydrochemical Features and Variabillity of Forest-Steppe Lakes in the Southwest of Western Siberia

While discussing the lakes in the southwest of Siberia, we noted that differences between neighboring groups are often less pronounced than the intergroup variability. The main tendencies are related to the changes in geographical context—first of all, to the distance from the Ural Mountains, which is pronounced as a decrease in TDS and the transformation of water type. The fact that the waters of the group V lakes are of the magnesium bicarbonate type, while all other groups correspond to the sodium chloride type, can be explained by the difference in geomorphological context. The V group lakes occur in the central part of the Ishim Plain at a significant distance from the large river valley. Such context predetermines thicker cover of late quaternary loess-like loams and high heterogeneity of covering deposits and, therefore, different hydrogeological conditions. Unlike the I-IV localities, salinization of soils is less common for the central part of the Ishim Plain in contrast with the Trans-Ural Plain [48].

The intergroup variability of main hydrological parameters for the studied lakes was rather high. In the case of groups I and III, this fact can be explained by the presence of

water bodies significantly different in terms of TDS, as well as adjacent lakes that can form gradients from mineralized to brine-type aquatic environments. In fact, the differences in TDS within one group of water bodies can exceed three orders of magnitude (Figure 3; Table A5 of the Appendix A). It is possible that such drastic differences stem from variable evaporation regime, leading to intensive temporal fluctuations of the water level and chemistry of the water column [49–52]. The presence of brine-type waters within the group I may also explain strong differences in the DOC and DIC concentrations.

The lakes of group II and V are characterized by stronger fluctuations in pH, probably related to macrophytes that overgrow the water reservoir as known in other settings [53]. Nevertheless, overall variability of pH values, both within each group and in the entire sample set, was low (CV < 33%), which implies a homogeneity of acid-base conditions (Table A5). According to the Kruskal–Wallis and the median tests (Figure 3), there were no statistically significant differences between the groups of lakes in terms of the median pH value.

There are multiple reason of high intergroup hydrochemical variability of the studied forest-steppe water bodies. Each group is characterized by a similar morphology, genesis and general landscape context. The statistical treatment of the data revealed weak positive correlation between the surface area of the studied lakes and concentration of anions and cations (Table A6 of the Appendix A). Tentatively, this can be explained by the variations in water chemistry as a result of different degree of evaporation and groundwater influence. The more evident reason for high intergroup variability of water chemistry within the certain group is the context of late quaternary evolution of aquatic environments and adjacent area. The morphology of shoreline, as well as adjacent lake depression in most cases shows signs of strong long-term level fluctuations [54–56], leading to connectivity of water bodies through recently dry channels. After the isolation of single lakes, their further evolution could be controlled by different local erosional activity within the watershed and potential of evaporation which, in turn, are related to the morphology of specific depression.

When comparing the studied lakes with water bodies of other semiarid regions of Western Siberia, it is worth mentioning, that they were characterized by sodium chloride waters, and rarely by magnesium bicarbonate type, while within the Baraba lowland and Kulunda plain bicarbonate-sodium water composition was more common [7]. An increase in evaporation and a decrease in precipitation in the Central Eurasian region, including south of Western Siberia as a results of on-going climate warming will likely increase overall TDS and lead to preferential removal of $CaCO_3$ and, in a lesser degree, $MgCO_3$-rich carbonates from the water column to the sediments. This may lead to further enrichment of the lake waters in Na, Cl, and $SO_4$ while impoverishing the waters in Ca, DIC and Mg. Such a transformation will certainly decrease the lake productivity, water quality and overall utility of the lakes for ecosystem services.

Another characteristic feature of the studied lakes is related to rather high DOC values. It is generally accepted that DOC values should decrease, while DIC increase across the aridization gradient. In our case the average DOC values for all mentioned water bodies reach 80 mg $L^{-1}$, which is comparable or higher than that in arctic and boreal thermokarst lakes of permafrost peatlands, Western Siberia [57] and higher than that in semiarid and arid lakes of Chinese regions [58,59]. On the other hand, similar to the Western Siberian forest-steppe lakes, high DOC values (32–330 mg $L^{-1}$) are reported for the water bodies of Alberta, Canada [60]. In the case of Canada, this fact is explained by an increase in DOC concentration with an increase in water residence time, leading to sizable evapo-concentration with increasing salinity, while preserving the internal DOC sources [60,61]. Therefore, a high DOC level in the studied lakes of southwestern Siberia can be viewed as a typical feature of the transitional ecotone of steppe/forest-steppe, analogous to that in Northern America. Note that the bottom sediments of lakes within the Ishim Plain are represented by predominantly organic-rich sapropels [19,26].

*4.2. Diversity and Factors of Carbonate Formation in Bottom Sediments of Lakes in the Southwest of Western Siberia*

A high Mg/Ca ratio is considered among the most important factors of endogenous Mg-Ca and Mg-carbonate mineral formation [62–65], whereas high values of (Na+K)/(Ca+Mg) molar ratio are an indicator of calcite precipitation [17]. Figure 9 illustrates the interrelations between the Mg/Ca ratio and the predominant associations of carbonate minerals.

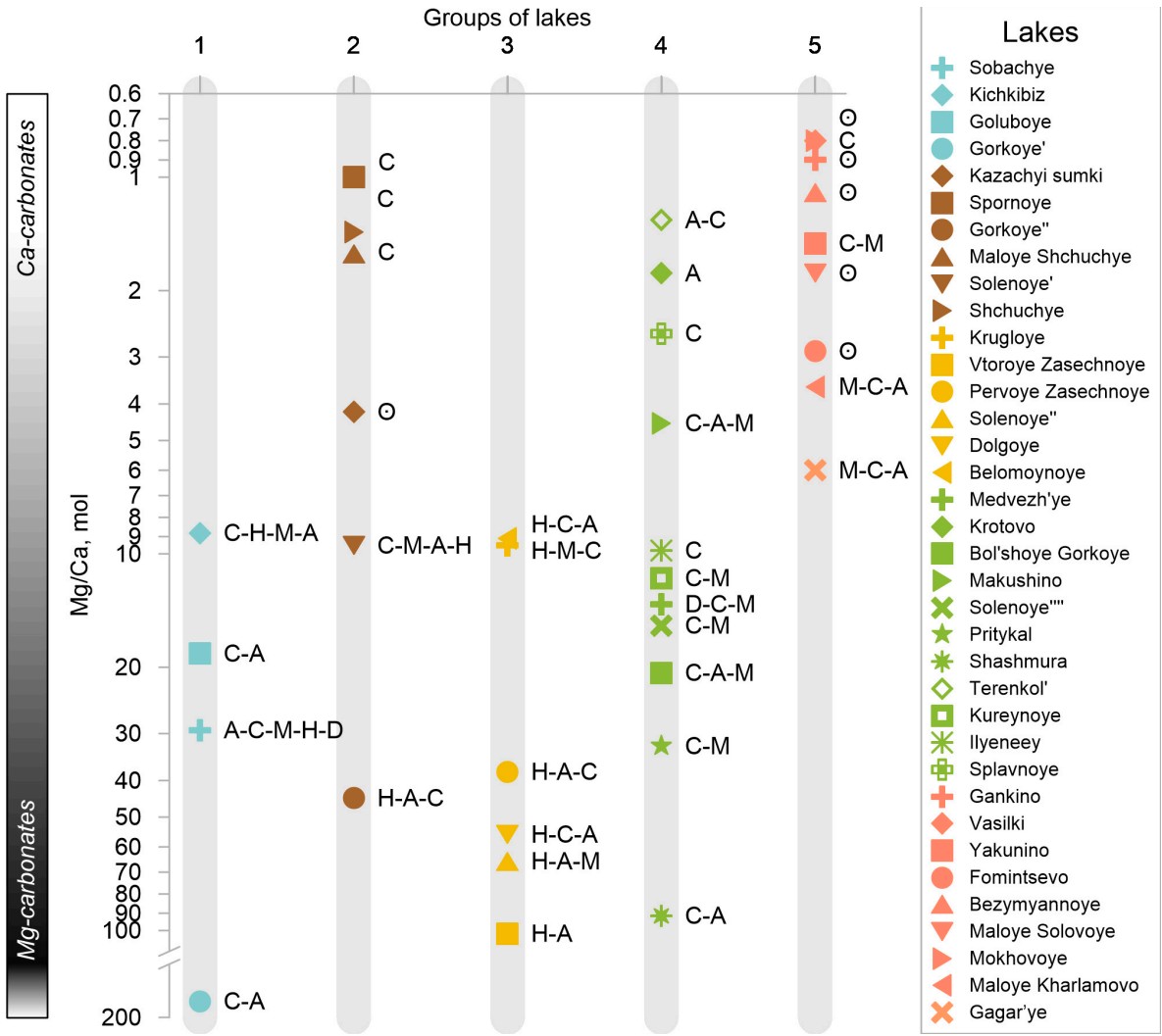

**Figure 9.** Relationship between Mg/Ca molar ratio in lake waters and authigenic carbonate mineralogy in sediments. Carbonates: C calcite, A aragonite, M Mg-calcite, D dolomite, G hydromagnesite, and ☉ not found.

Thus, analyzing Figure 9, we can conclude that the authigenic carbonates in bottom sediments of the studied lakes are represented predominantly by Ca-carbonates with a Mg/Ca ratio in water below 32. Mg-Ca-carbonates appear at a Mg/Ca ratio ranging from 1.5 to 13.6, Mg-carbonates appear at an Mg/Ca of 9.5 and predominate in the composition at a Mg/Ca ratio between 44.5 and 102. The studied lakes meet all these criteria and hence are capable of precipitating various Ca, Mg carbonates in the water column and sediments. According to the results of thermodynamic assessment of SI (Figure 5; Table A3), not all of the above carbonates that could potentially precipitate in the studied aquatic environments were observed. At the same time, the results of XRD analysis for all studied lakes (Table A4 of the Appendix A), as well as hydrochemical studies (Table A2), make it possible to assess the influence of various factors on the occurrence of most common carbonates.

Thus, aragonite is present in bottom sediments containing gastropod shells of anthemia cists, which are abundant in bottom sediments of some water bodies (all lakes of the I and III groups and certain water bodies from other groups). This may reflect some dependence of the degree eutrophication (or simply overgrowth by macrophytes) of lakes and the amount of aragonite in bottom sediment samples. Overall, observations on aragonite occurrences in the studied lakes corroborate the importance of aragonite in mineralization of skeletal structures reported in former studies [66,67].

Calcite is found almost everywhere; its content varies significantly depending on the composition of the waters and the granulometric composition of bottom sediments. Hydromagnesite is typical only for a number of water bodies of I and III groups with very high Mg/Ca ratios. The localization and the morphology of aggregates in general was rather similar to other reported case of modern hydromagnesite within the aquatic environments of mineralized lakes and lagoons [68–72]. The hydromagnesite observed in the studied water bodies is often associated with biogenic processes, as well as for a number of reported cases [73–75]. Dolomite, which is formed in some reservoirs of III and IV key areas, is mostly observed using SEM, since it is not always identified on X-ray diffraction analysis due to its low content. Mg-calcite is observed in varying amounts in water bodies of all key areas. This mineral mainly forms aggregates with hydromagnesite or dolomite, and presents as large communities of single elongated split sheaf-like microcrystals. These crystals likely occur within EPS films produced by bottom microbial consortia, and therefore rarely forms as large poorly cemented homogeneous nodules.

Note that in this work, we could not assess the impact of anthropogenic activity on lake hydrochemistry and the carbon cycle. In fact, such a study would require a comprehensive analysis of major nutrients (N, P) in the water column and incoming streams. We would expect here that different forms of dissolved nitrogen originated form fertilizers or sewage would not directly impact carbonate mineral formation, other than increasing the frequency of bloom or coastal biofilm growth. In contrast, excess phosphate delivery from the watershed may not only boost the algal activity, but directly inhibit $CaCO_3$ precipitation in the water column, while affecting hydrous $MgCO_3$ precipitation in a lesser degree.

In general, our study demonstrate that Mg-carbonate formation is potentially important for assessment of carbon sequestration within the studied aquatic environments, given that this process was reported for at least two groups of lakes present in this study. Taking into account the fact that Mg-carbonates were also reported for a number of other lakes of south Siberia and Central Eurasia by D.S Shlyapnikov [26], we can suggest that the process of their formation is not a very rare case for the studied part of Western Siberia. Unlike the Baraba and Kulunda regions were precipitation of Mg-Ca and Ca carbonates was predominant [7,76,77], in our study the Mg-carbonates significantly contributed to the authigenic mineral formation. Therefore, estimation of the C burial flux related to this process, understanding the relative role of biotic and abiotic controls, paleo-limnological aspect, as well as modern abundance of Mg-rich carbonate should be the priority of more comprehensive studies. These should allow enable estimation of the potential of south-western West Siberian water bodies, including saline lakes, in C storage and inorganic C burial under the natural and anthropogenically induced transformation of climate and landscapes [78].

## 5. Conclusions

The present study reports new data on the hydrochemistry of the water column and bottom sediments mineralogy of drainless lakes in southwestern Siberia in relation to the inorganic C cycle and $CO_2$ uptake/release by these lakes. Three main conclusions are drawn:

- Lake waters exhibited significant differences in hydrochemistry, with sizable heterogeneity among groups of lakes and among water bodies within each group. Most lakes have weakly alkaline to strongly alkaline waters and are of the saline or brine type. A weak correlation was found between the morphology of lakes and the composition of

waters. According to the composition of major solutes, most lakes are of the sodium-potassium chloride-sulfate salt type. The lakes of the eastern V group, confined to the Ishim Plain, are distinguished by the highest content of organic matter in bottom sediments and sizable amount of aquatic plants (macrophytes) in the coastal zones. The water bodies of the III group are characterized by the highest content of DOC and the lowest Ca concentrations in the water column. The lakes of the third group are distinguished from the other groups by the highest values of the Mg/Ca ratio. Overall, high DOC concentrations are characteristic for all studied groups of lakes.

- The mineral composition of lake sediments, as determined by XRD and SEM-EDS analyses, demonstrated that the bulk of sediments is represented by a terrigenous component, consisting mainly of quartz and feldspars. Among authigenic minerals, minerals of the carbonate class are most common, and to a lesser extent, halides, sulfates, sulfides, oxides, and hydroxides. Heterogeneity of the mineral composition, primarily in terms of authigenic minerals, was observed both between groups, and within certain water bodies. Alkaline-earth carbonates are the most common class of authigenic minerals and include calcite, Mg-calcite, aragonite, Ca-dolomite and hydromagnesite, as well as siderite and rhodochrosite in some cases.

- Water bodies of the group III with highest Mg/Ca molar ratios are characterized by the greatest variety of magnesium minerals, primarily magnesium carbonates, and also have abundant signs of biomineralization, whereas bottom sediments of the group V are distinguished by the presence of iron oxides, which, apparently, can be explained by the lithology and the nature of land use of adjacent agricultural areas within the lake watershed. Overall, the observed differences in the composition of the authigenic part of the mineral component of sediments may stem from a combination of factors, including, but not limited to, the degree of their eutrophication, the intensity of the erosion and agricultural activity within the watershed.

- The main limitation of the present study is its low seasonal resolution (lack of spring, autumn and winter measurements) when extensive exchange of $CO_2$ with atmosphere could occur, as known from works on other regions in Siberia [79]. The obtained results, however, encompass the most active open water period and, as such, can be considered as representative for assessing atmospheric C sequestration potential in bottom sediments. Missing in these evaluations is organic C burial [80], which should be a focus of further research.

**Author Contributions:** Conceptualization, A.K. (Alexandr Konstantinov) and O.S.P.; methodology, A.N., A.K. (Alexandr Konstantinov) and O.S.P.; validation, A.K. (Alexandr Konstantinov) and E.K.; formal analysis, A.N. and E.K.; investigation, A.N., A.K. (Alexandr Konstantinov), E.K., Y.S., A.L. and A.K. (Alina Kurasova); resources, S.L.; data curation, A.N. and E.K.; writing—original draft preparation, A.N., A.K. (Alexandr Konstantinov) and E.K.; writing—review and editing, A.K. (Alexandr Konstantinov), E.K. and O.S.P.; visualization, A.N., E.K. and Y.S.; supervision, A.K. (Alexandr Konstantinov) and O.S.P.; project administration, A.K. (Alexandr Konstantinov); funding acquisition, O.S.P. All authors have read and agreed to the published version of the manuscript.

**Funding:** O.S.P. is grateful for support from the TSU Development Programme PRIORITY—2030 and ANR MeLiCa.

**Data Availability Statement:** Data are openly available within this article in Appendix A.

**Conflicts of Interest:** The authors declare no conflict of interest.

## Appendix A

**Table A1.** Location and morphological parameters of the studied lakes.

| Lake | Group | Coordinates | | Area, km² | Watershed, km² | Average Depth, m |
|---|---|---|---|---|---|---|
| Sobachye | I | 54°22′4.97″ | 61°40′42.01″ | 0.45 | 1.57 | 1.8 |
| Kichkibiz | I | 54°20′6.12″ | 61°40′25.07″ | 5.96 | 12.4 | 2.5 |
| Goluboye | I | 54°20′1.73″ | 61°45′0.36″ | 1.4 | 2.9 | 2 |
| Gorkoye′ | I | 54°19′22.49″ | 61°43′4.73″ | 3.63 | 5.9 | 1.5 |
| Kazachka | II | 54°16′59.34″ | 62°31′7.55″ | 0.46 | 1.37 | 1.8 |
| Kazachyi Sumki | II | 54°18′8.61″ | 62°31′8.11″ | 0.65 | 1.12 | 1.6 |
| Spornoye | II | 54°19′56.78″ | 62°28′53.81″ | 0.41 | 0.75 | 1.3 |
| Gorkoye″ | II | 54°18′29.04″ | 62°27′57.05″ | 1 | 1.66 | 1.3 |
| Maloye Shchuchye | II | 54°19′27.95″ | 62°27′27.52″ | 0.4 | 0.92 | 1.6 |
| Solenoye′ | II | 54°18′34.18″ | 62°25′45.19″ | 0.27 | 0.6 | 1.4 |
| Shchuchye | II | 54°19′48.53″ | 62°26′5.51″ | 0.5 | 0.87 | 1.5 |
| Krugloye | III | 54°37′40.93″ | 64°1′42.36″ | 1.27 | 2.62 | 1.5 |
| Lomovo | III | 54°37′40.93″ | 64°1′42.36″ | 0.67 | 1.67 | 1.4 |
| Vtoroye Zasechnoye | III | 54°37′42.85″ | 64°0′49.08″ | 1.68 | 3.62 | 2 |
| Pervoye Zasechnoye | III | 54°37′40.93″ | 64°1′42.36″ | 1.39 | 3.14 | 1.1 |
| Solenoye″ | III | 54°37′40.93″ | 64°1′42.36″ | 1.11 | 2.44 | 2.3 |
| Dolgoye | III | 54°38′27.99″ | 64°1′33.74″ | 1.7 | 4.51 | 1.8 |
| Labzovitoye | III | 54°39′29.48″ | 63°55′28.94″ | 0.22 | 0.4 | 1.7 |
| Belomoynoye | III | 54°39′34.36″ | 63°53′21.08″ | 1.25 | 1.97 | 1.5 |
| Medvezh'ye | IV | 55°11′30.70″ | 67°57′5.77″ | 67.3 | 161 | 0.6 |
| Krotovo | IV | 55°11′37.68″ | 66°54′48.40″ | 0.54 | 1.51 | 1.1 |
| Bol'shoye Gorkoye | IV | 55°11′58.91″ | 66°53′23.57″ | 3.12 | 5.31 | 1.2 |
| Solenoye‴ | IV | 55°8′52.62″ | 66°54′47.29″ | 7.25 | 12.9 | 1.6 |
| Krutoyar | IV | 55°9′49.59″ | 66°57′25.61″ | 0.47 | 0.78 | 1.4 |
| Zolotoye | IV | 55°6′43.99″ | 66°58′42.62″ | 1.08 | 2.44 | 2.3 |
| Makushino | IV | 55°12′0.01″ | 67°14′7.10″ | 2.45 | 4.21 | 1.8 |
| Gorkoye‴ | IV | 55°4′40.25″ | 67°1′39.21″ | 0.86 | 1.4 | 1.7 |
| Solenoye⁗ | IV | 55°2′53.15″ | 66°59′17.40″ | 0.83 | 1.35 | 1.5 |
| Pritykal | IV | 54°58′26.95″ | 66°58′39.04″ | 0.44 | 0.63 | 1.6 |
| Shashmura | IV | 54°57′10.48″ | 66°58′9.67″ | 0.62 | 0.95 | 1.1 |
| Terenkol′ | IV | 54°55′18.74″ | 66°56′40.71″ | 2.7 | 4.2 | 1.2 |
| Kureynoye | IV | 54°54′29.73″ | 66°57′13.31″ | 3.12 | 5.73 | 1.3 |
| Ilyeneey | IV | 55°2′35.91″ | 66°55′36.77″ | 3.57 | 5.47 | 1.3 |
| Splavnoye | IV | 66°55′36.77″ | 66°58′57.44″ | 1.67 | 3.32 | 1.2 |
| Gankino | V | 55°49′27.39″ | 67°58′25.80″ | 0.37 | 0.72 | 1.6 |
| Vasilki | V | 55°48′48.33″ | 67°58′23.01″ | 0.42 | 0.68 | 1.5 |
| Yakunino | V | 55°47′47.18″ | 68°0′48.97″ | 0.75 | 1.25 | 2 |
| Fomintsevo | V | 55°56′26.29″ | 67°52′49.61″ | 1.37 | 2.01 | 2.1 |
| Bezymyannoye | V | 55°56′47.33″ | 67°55′7.17″ | 0.33 | 1.13 | 1.9 |
| Maloye Solovoye | V | 55°56′49.92″ | 67°59′45.33″ | 2.21 | 3.66 | 3 |
| Mokhovoye | V | 55°55′32.18″ | 68°0′9.00″ | 0.32 | 0.55 | 2.5 |
| Maloye Kharlamovo | V | 55°54′2.02″ | 67°43′12.71″ | 1.43 | 2 | 3.1 |
| Gagar'ye | V | 55°56′43.00″ | 67°45′41.41″ | 0.38 | 1.1 | 1.6 |

**Table A2.** Physical and hydrochemical parameters of the studied lakes.

| Lake | Date of Sampling | T, °C | pH | TDS | Mg²⁺ | Ca²⁺ | Na⁺ | K⁺ | Cl⁻ | SO₄²⁻ | DOC | DIC | Mg/Ca | (Na+K)/(Ca+Mg) |
|---|---|---|---|---|---|---|---|---|---|---|---|---|---|---|
| | | | | g L⁻¹ | | | | mg L⁻¹ | | | | | | mol |
| Sobachye | 08.08.2019 | 22.3 | 9.01 | 21.6 | 1323.92 | 74.22 | 6915.42 | 209.19 | 5607.41 | 3630.65 | 119.03 | 193.7 | 29.4 | 5.4 |
| Kichkibiz | 08.08.2019 | 20.8 | 8.93 | 23.0 | 1483.75 | 278.22 | 7150.81 | 136.4 | 6613 | 3359 | 56 | 69 | 8.8 | 4.6 |
| Goluboye | 08.08.2019 | 18.7 | 9.15 | 9.84 | 555.71 | 49.68 | 3557.47 | 90.84 | 2060 | 1452.1 | 40.28 | 224.09 | 18.4 | 6.5 |
| Gorkoye' | 09.08.2019 | 23.1 | 7.7 | 122 | 11,165.43 | 92.98 | 55,617.87 | 438.69 | 83,090.75 | 63,892.74 | 1226 | 493.58 | 198 | 5.3 |
| Kazachka | 04.08.2019 | 20.8 | 7.9 | 0.33 | 12.04 | 22.48 | 20.05 | 13.89 | 28.26 | 3.78 | 18.25 | 29.1 | 0.9 | 1.2 |
| Kazachyi sumki | 04.08.2019 | 20.2 | 8.9 | 7.07 | 233.39 | 91.01 | 1300.4 | 37.75 | 2266.02 | 50.11 | 25.31 | 67.75 | 4.2 | 4.8 |
| Spornoye | 05.08.2019 | 20.5 | 9.5 | 0.49 | 11.71 | 18.42 | 59.98 | 27.25 | 77.43 | 8 | 24.23 | 24.97 | 1 | 3.5 |
| Gorkoye'' | 05.08.2019 | 24.8 | 8.9 | 41.5 | 2217.79 | 82.13 | 10,786.64 | 143.77 | 15,898.1 | 557.29 | 83.56 | 110 | 44.5 | 5.1 |
| Maloye Shchuchye | 05.08.2019 | 21.6 | 10.4 | 0.47 | 7.34 | 7.78 | 89.32 | 11.55 | 75.97 | 1 | 23.77 | 20.48 | 1.6 | 8.4 |
| Solenoye' | 06.08.2019 | 23.9 | 7.8 | 26.7 | 1308.08 | 226.26 | 6469.91 | 75.82 | 8784.08 | 1823.24 | 68.24 | 135.09 | 9.5 | 4.8 |
| Shchuchye | 06.08.2019 | 23.8 | 10.1 | 0.53 | 19.05 | 21.75 | 87 | 14.34 | 86.35 | 3.7 | 18.28 | 27.43 | 1.4 | 3.1 |
| Krugloye | 28.09.2018 | 22.3 | 9.12 | 42.4 | 170.49 | 29.71 | 9126.53 | 157.9 | 13,568.32 | 308.93 | 112.08 | 343.07 | 9.5 | 51.7 |
| Lomovo | 28.09.2018 | 12.1 | 9.22 | 45.1 | 642.59 | 23.28 | 11,225.81 | 139.1 | 11,630.54 | 8731.11 | 102 | 364.91 | 45.5 | 18.2 |
| Vtoroye Zasechnoye | 28.09.2018 | 24.3 | 9.1 | 67.4 | 1391.16 | 22.48 | 15,466.77 | 188.31 | 22,136 | 5645 | 95 | 275 | 102 | 11.7 |
| Pervoye Zasechnoye | 29.09.2018 | 23 | 8.79 | 41.6 | 826.78 | 35.94 | 8629.57 | 110.53 | 12,605.69 | 3132.77 | 68.82 | 271.67 | 37.9 | 10.8 |
| Solenoye'' | 29.09.2018 | 24.9 | 8.84 | 84 | 2076.81 | 52.27 | 19,954.63 | 272.92 | 27,896 | 8754 | 100 | 240 | 65.5 | 10.1 |
| Dolgoye | 30.09.2018 | 21 | 9.63 | 29.5 | 248.72 | 7.38 | 6167.87 | 119.98 | 8809.44 | 330.4 | 88.42 | 354.62 | 55.6 | 26.0 |
| Labzovitoye | 30.09.2018 | 9.6 | 7.53 | 0.75 | 17.14 | 29.54 | 98.41 | 7.53 | 597.48 | 19.77 | 23.52 | 81.08 | 1 | 3.1 |
| Belomoynoye | 30.09.2018 | 12.7 | 9.51 | 8.01 | 82.74 | 15.03 | 1761.57 | 121.95 | 1390.12 | 642.22 | 57.4 | 377.16 | 9.1 | 21.1 |
| Medvezh'ye | 07.08.2018 | 23.4 | 7.2 | 160 | 3851.12 | 467.92 | 17,505.08 | 50.95 | 44,010.86 | 10,014 | 97.8 | 52.26 | 13.6 | 4.5 |
| Krotovo | 07.08.2018 | 13.4 | 8.69 | 0.15 | 47.64 | 43.18 | 213.45 | 15.02 | 215.07 | 79.1 | 11.86 | 54.8 | 1.8 | 3.2 |
| Bol'shoye Gorkoye | 07.08.2018 | 13.6 | 8.41 | 53.1 | 2532.57 | 201.84 | 9751.27 | 88.91 | 17,768.63 | 6751.59 | 71.04 | 93.72 | 20.7 | 3.9 |
| Solenoye''' | 07.08.2018 | 14.2 | 8.39 | 75.6 | 4018.55 | 276.04 | 14,721.92 | 73.16 | 26,705.89 | 11,233 | 73.44 | 101.28 | 2 | 3.7 |
| Krutoyar | 07.08.2018 | 15.0 | 8.91 | 14.5 | 283.55 | 54.39 | 2933.92 | 37.51 | 3439.42 | 2002.95 | 47.06 | 195.23 | 8.6 | 9.9 |
| Zolotoye | 18.03.2019 | 0.9 | 9.1 | 6.4 | 144.75 | 126.75 | 524.55 | 129.17 | 742.61 | 13.27 | 73.48 | 190.57 | 1.9 | 2.9 |
| Makushino | 07.08.2019 | 18 | 8.24 | 2.7 | 127.78 | 46.71 | 511.15 | 46.15 | 506.49 | 124.78 | 25.42 | 125.12 | 4.5 | 3.6 |
| Gorkoye''' | 07.08.2019 | 21.6 | 8.56 | 9.86 | 500.42 | 163.92 | 2047.64 | 35.59 | 2677.86 | 1086.42 | 26.12 | 103.09 | 5. | 3.6 |
| Solenoye'''' | 07.08.2019 | 22 | 8.35 | 59.1 | 5268.86 | 560.52 | 16,660.18 | 87.7 | 21,319.44 | 6668.22 | 64.35 | 75.17 | 15.5 | 3.1 |
| Pritykal | 07.08.2019 | 22 | 9.04 | 22.6 | 889.74 | 45.22 | 8138.55 | 92.95 | 5890.3 | 3320.28 | 69.83 | 282.22 | 32.4 | 9.4 |
| Shashmura | 07.08.2019 | 22.5 | 8.76 | 105 | 7538.47 | 136.03 | 47,308.13 | 176.2 | 37,500.75 | 22,717.09 | 160.34 | 211.21 | 91.4 | 6.6 |
| Terenkol' | 08.08.2019 | 20.3 | 9.26 | 0.7 | 26.62 | 34.53 | 112.64 | 22.97 | 155.7 | 32.18 | 12.37 | 43.29 | 1.3 | 2.8 |
| Kureynoye | 08.08.2019 | 20.5 | 8.94 | 13.7 | 649.15 | 92.04 | 3980.9 | 129.14 | 3911.3 | 960.93 | 50.99 | 172.97 | 11.6 | 6.1 |
| Ilyeneey | 08.08.2019 | 20 | 7.9 | 49.1 | 3988.41 | 668.58 | 13,587.42 | 161.31 | 17,750.18 | 4837.43 | 70.66 | 77.74 | 9.8 | 3.3 |
| Splavnoye | 09.08.2019 | 19.9 | 8.92 | 2.1 | 97 | 61.55 | 353.45 | 25.86 | 404.79 | 141.96 | 20.99 | 93.8 | 2.6 | 2.9 |
| Gankino | 01.08.2019 | 16.7 | 9.6 | 0.25 | 5.87 | 10.33 | 25.91 | 13.05 | 23.34 | 4.01 | 16.19 | 19.97 | 0.9 | 2.9 |
| Vasilki | 01.08.2019 | 16.8 | 7.56 | 0.44 | 11.4 | 23.76 | 51.36 | 19.34 | 53.06 | 0.32 | 18.85 | 36.82 | 0.8 | 2.6 |
| Yakunino | 01.08.2019 | 17.6 | 8.22 | 0.83 | 30.01 | 34.12 | 117.16 | 17.47 | 89.55 | 12.29 | 26.94 | 75.27 | 1.5 | 2.7 |
| Fomintsevo | 02.08.2019 | 19.9 | 9.63 | 0.97 | 28.65 | 16.1 | 167.54 | 15.61 | 116.04 | 22.9 | 28.48 | 73.11 | 2.9 | 4.9 |
| Bezymyannoye | 03.08.2019 | 19.4 | 8.5 | 0.58 | 14.15 | 20.77 | 116.02 | 9.91 | 12.31 | 1.41 | 28.64 | 72.75 | 1.1 | 4.8 |
| Maloye Solovoye | 03.08.2019 | 18.4 | 6.9 | 0.11 | 50.05 | 46.87 | 207.51 | 20.72 | 347.28 | 2.41 | 20.7 | 57.79 | 1.8 | 3.0 |
| Mokhovoye | 03.08.2019 | 18.5 | 9.01 | 0.34 | 11.82 | 24.09 | 28.09 | 22.53 | 8.58 | 0.29 | 18.61 | 39.91 | 0.8 | 1.7 |
| Maloye Kharlamovo | 03.08.2019 | 18.0 | 9.1 | 3.83 | 113.96 | 52.38 | 585.71 | 19.34 | 729.73 | 21.88 | 32.41 | 105.23 | 3.6 | 4.3 |
| Gagar'ye | 04.08.2019 | 17.6 | 7.4 | 3.07 | 200.98 | 55.45 | 905.60 | 22.11 | 1142.97 | 224.32 | 34.38 | 116.01 | 6.0 | 4.1 |

**Table A3.** Saturation indices for selected minerals, calculated with Visual MINTEQ for the studied lakes.

| Lake | Arg | Art | MHC | C | DD | OD | H | HM | M | V |
|---|---|---|---|---|---|---|---|---|---|---|
| Sobachye | 1.305 | 0.689 | 0.105 | 1.450 | 3.932 | 4.493 | 6.224 | 2.337 | 1.973 | 0.877 |
| Kichkibiz | 1.380 | 0.030 | 0.180 | 1.520 | 3.500 | 4.070 | 4.800 | 0.180 | 1.520 | 0.950 |
| Goluboye | 1.422 | 0.385 | 0.225 | 1.570 | 3.921 | 4.497 | 5.982 | 1.675 | 1.968 | 0.987 |
| Gorkoye' | 0.216 | −1.735 | −1.007 | 0.361 | 2.592 | 3.150 | 4.376 | −0.871 | 1.695 | −0.211 |
| Kazachka | −0.243 | −6.117 | −1.438 | −0.096 | −0.704 | −0.136 | −4.577 | −13.719 | −1.064 | −0.674 |
| Kazachyi sumki | 1.262 | −1.049 | 0.066 | 1.409 | 2.951 | 3.521 | 3.382 | −2.198 | 1.107 | 0.830 |
| Spornoye | 0.997 | −1.656 | −0.198 | 1.144 | 1.893 | 2.462 | 0.736 | −5.184 | 0.303 | 0.565 |
| Gorkoye'' | 1.044 | 0.708 | −0.156 | 1.188 | 3.566 | 4.117 | 5.632 | 2.022 | 1.785 | 0.621 |
| Maloye Shchuchye | 0.952 | 0.106 | −0.243 | 1.098 | 2.059 | 2.623 | 1.315 | −2.798 | 0.477 | 0.522 |
| Solenoye' | 0.673 | −2.756 | −0.526 | 0.817 | 2.166 | 2.721 | 2.180 | −4.521 | 0.786 | 0.248 |
| Shchuchye | 1.409 | 0.457 | 0.216 | 1.553 | 2.932 | 3.487 | 3.006 | −1.213 | 0.819 | 0.984 |
| Krugloye | 1.263 | −0.534 | 0.063 | 1.409 | 3.373 | 3.934 | 4.631 | −0.439 | 1.456 | 0.835 |
| Lomovo | 0.958 | −0.194 | −0.238 | 1.111 | 3.261 | 3.866 | 4.996 | 0.575 | 2.015 | 0.509 |
| Vtoroye Zasechnoye | 0.906 | 0.989 | −0.297 | 1.050 | 3.676 | 4.229 | 6.241 | 3.047 | 2.049 | 0.482 |
| Pervoye Zasechnoye | 0.987 | −0.195 | −0.213 | 1.132 | 3.398 | 3.956 | 5.254 | 0.799 | 1.734 | 0.560 |
| Solenoye'' | 1.006 | 0.544 | −0.198 | 1.150 | 3.681 | 4.231 | 6.051 | 2.310 | 1.934 | 0.584 |
| Dolgoye | 0.938 | 1.050 | −0.261 | 1.085 | 3.514 | 4.080 | 5.712 | 2.553 | 1.966 | 0.508 |
| Labzovitoye | −0.384 | −7.731 | −1.566 | −0.228 | −1.173 | −0.557 | −5.598 | −16.202 | −0.983 | −0.837 |
| Belomoynoye | 1.308 | −0.487 | 0.118 | 1.461 | 3.373 | 3.975 | 4.622 | −0.437 | 1.753 | 0.861 |
| Medvezh'ye | −0.242 | −4.596 | −1.451 | −0.097 | 0.438 | 0.995 | −1.170 | −8.806 | −0.010 | −0.668 |
| Krotovo | 0.830 | −3.207 | −3.207 | 0.983 | 1.611 | 2.210 | 0.286 | −7.014 | 0.443 | 0.385 |
| Bol'shoye Gorkoye | 0.754 | −1.660 | −0.446 | 0.906 | 2.481 | 3.079 | 3.046 | −2.640 | 1.382 | 0.308 |
| Solenoye''' | 0.836 | −1.384 | −0.367 | 0.988 | 2.714 | 3.309 | 3.574 | −1.922 | 1.510 | 0.392 |
| Krutoyar | 1.189 | −1.234 | −0.005 | 1.340 | 3.049 | 3.641 | 3.867 | −1.823 | 1.463 | 0.747 |
| Zolotoye | 1.796 | −1.978 | 0.642 | 1.958 | 3.304 | 3.960 | 3.595 | −3.413 | 1.672 | 1.323 |
| Makushino | 0.730 | −3.157 | −0.465 | 0.879 | 1.893 | 2.472 | 1.288 | −5.866 | 0.657 | 0.293 |
| Gorkoye''' | 1.265 | −1.334 | 0.069 | 1.412 | 3.060 | 3.624 | 3.692 | −2.176 | 1.164 | 0.836 |
| Solenoye'''' | 1.120 | −0.625 | −0.087 | 1.266 | 3.217 | 3.779 | 4.451 | −0.575 | 1.453 | 0.691 |
| Pritykal | 1.278 | 0.563 | 0.079 | 1.424 | 3.922 | 4.484 | 6.250 | 2.263 | 2.000 | 0.850 |
| Shashmura | 1.115 | 1.067 | −0.101 | 1.261 | 3.993 | 4.553 | 6.785 | 3.447 | 2.217 | 0.688 |

**Table A3.** *Cont.*

| Lake | Arg | Art | MHC | C | DD | OD | H | HM | M | V |
|---|---|---|---|---|---|---|---|---|---|---|
| Terenkol' | 1.259 | −1.525 | 0.064 | 1.407 | 2.466 | 3.036 | 1.932 | −4.120 | 0.621 | 0.828 |
| Kureynoye | 1.455 | −0.031 | 0.258 | 1.602 | 3.793 | 4.361 | 5.518 | 0.761 | 1.745 | 1.024 |
| Ilyeneey | 0.833 | −2.238 | −0.373 | 0.980 | 2.417 | 2.988 | 2.640 | −3.710 | 1.009 | 0.400 |
| Splavnoye | 1.393 | −1.342 | 0.198 | 1.541 | 3.029 | 3.600 | 3.356 | −2.647 | 1.064 | 0.960 |
| Gankino | 0.739 | −2.278 | −0.455 | 0.889 | 1.267 | 1.851 | −0.597 | −6.880 | 0.069 | 0.300 |
| Vasilki | −0.549 | −7.464 | −1.743 | −0.399 | −1.435 | −0.851 | −6.127 | −16.309 | −1.349 | −0.988 |
| Yakunino | 0.519 | −4.400 | −0.676 | 0.668 | 0.983 | 1.564 | −1.014 | −9.200 | −0.027 | 0.081 |
| Fomintsevo | 1.278 | −0.415 | 0.083 | 1.425 | 2.941 | 3.512 | 3.322 | −1.638 | 1.091 | 0.845 |
| Bezymyannoye | 0.598 | −4.029 | −0.597 | 0.746 | 1.089 | 1.662 | −0.872 | −8.767 | −0.064 | 0.165 |
| Maloye Solovoye | −0.875 | −8.124 | −2.070 | −0.726 | −1.728 | −1.151 | −6.368 | −16.885 | −1.373 | −1.311 |
| Mokhovoye | 0.924 | −2.943 | −0.270 | 1.073 | 1.574 | 2.151 | −0.060 | −7.195 | 0.127 | 0.489 |
| Maloye Kharlamovo | 1.453 | −0.862 | 0.258 | 1.602 | 3.265 | 3.844 | 3.959 | −1.623 | 1.307 | 1.016 |
| Gagar'ye | −0.163 | −5.510 | −1.359 | −0.014 | 0.212 | 0.793 | −1.964 | −10.578 | −0.116 | −0.601 |

Notes: Arg Aragonite, Art Artinite, MHC Monohydrocalcite, C Calcite, DD Disordered dolomite, OD ordered dolomite, H Huntite, HM Hydromagnesite, M Magnesite, and V Vaterite.

**Table A4.** Mineralogical composition (%) of surface sediments of the studied lakes.

| Lake | Quartz | Calcite | Mg-Calcite | Feldspar | Gypsum | Aragonite | Dolomite | Illite | Hydromagnesite | Siderite | Goethite | Halite |
|---|---|---|---|---|---|---|---|---|---|---|---|---|
| Sobachye | 62.2 | 2.9 | 2 | 14.3 | – | 4.8 | 0.8 | 10.9 | 1.1 | – | – | 1 |
| Kichkibiz | 76 | 6.0 | 2.2 | 2.2 | 3.3 | 2 | – | 4.3 | 3.8 | – | – | 0.2 |
| Goluboye | 76.9 | 7.1 | – | 12.7 | – | 3.3 | – | – | – | – | – | – |
| Gorkoye' | 74.8 | 3.1 | – | 18.4 | 2.7 | 1 | – | – | – | – | – | – |
| Kazachyi sumki | 97.3 | – | – | 2.7 | – | – | – | – | – | – | – | – |
| Spornoye | 92.4 | 0.5 | – | 7.1 | – | – | – | – | – | – | – | – |
| Gorkoye'' | 57.9 | 2.1 | – | 8.5 | 1 | 2.5 | – | 22.4 | 5.6 | – | – | – |
| Maloye Shchuchye | 92.5 | 0.4 | – | 7.1 | – | – | – | – | – | – | – | – |
| Solenoye' | 34.7 | 12 | 6.1 | 5.6 | 1.4 | 5.6 | – | 20.4 | 5 | – | – | 9.2 |
| Shchuchye | 87 | 0.5 | – | 12.5 | – | – | – | – | – | – | – | – |
| Krugloye | 78.6 | 2.2 | 2.2 | 5.2 | 1.9 | – | – | 5 | 3 | – | – | 1.9 |
| Vtoroye Zasechnoye | 69.6 | – | – | 1.5 | – | 5.3 | – | 5.4 | 18.2 | – | – | – |
| Pervoye Zasechnoye | 81.9 | 0.1 | – | 2.8 | 2.3 | 2.7 | – | 7 | 2.8 | – | – | 0.4 |
| Solenoye'' | 78.1 | – | 1 | 2 | 2.4 | 1.7 | – | 6.4 | 4 | – | – | 4.4 |
| Dolgoye | 82.5 | 0.9 | – | 6.2 | 0.8 | 0.8 | 3 | 2.8 | 2.1 | – | – | 0.9 |
| Belomoynoye | 71.7 | 5.3 | – | 0.7 | 3.9 | 1.5 | 2.5 | 9.1 | 5.3 | – | – | – |
| Medvezh'ye | 75.5 | 2.5 | 1 | 9.7 | 3.7 | – | 3.6 | – | – | 4 | – | – |
| Krotovo | 72.9 | – | – | 25 | 0.7 | 1.4 | – | – | – | – | – | – |

**Table A4.** *Cont.*

| Lake | Quartz | Calcite | Mg-Calcite | Feldspar | Gypsum | Aragonite | Dolomite | Illite | Hydromagnesite | Siderite | Goethite | Halite |
|------|--------|---------|------------|----------|--------|-----------|----------|--------|----------------|----------|----------|--------|
| Bol'shoye Gorkoye | 59.1 | 10.0 | 1.8 | 17.1 | 4.6 | 7.4 | – | – | – | – | – | – |
| Makushino | 65.3 | 9.7 | 1.3 | 22.1 | – | 1.3 | – | – | – | – | – | 0.3 |
| Solenoye'''' | 69.8 | 6.6 | 0.2 | 20.8 | – | – | – | – | – | – | – | 2.6 |
| Pritykal | 78.9 | 1.3 | 0.5 | 18.5 | – | – | – | – | – | – | – | 0.8 |
| Shashmura | 53.8 | 24.2 | – | 16.2 | 2.2 | 1.3 | – | – | – | – | – | 2.3 |
| Terenkol' | 75.8 | 0.6 | – | 22.4 | – | 0.9 | – | – | – | – | – | 0.3 |
| Kureynoye | 57.5 | 14.5 | 2.4 | 17.1 | 5 | – | 0.6 | – | 2.9 | – | – | – |
| Ilyeneey | 94.3 | 1 | – | 3 | – | – | 1.7 | – | – | – | – | – |
| Splavnoye | 73.1 | 5 | – | 19.3 | 2.6 | – | – | – | – | – | – | – |
| Gankino | 74.9 | – | – | 25.1 | – | – | – | – | – | – | – | – |
| Vasilki | 53.5 | 0.8 | 7.8 | 6.1 | – | – | – | 31.8 | – | – | – | – |
| Yakunino | 29.4 | 10.5 | 12.4 | 5.8 | 5.4 | 4.5 | – | 32 | – | – | – | – |
| Fomintsevo | 53.7 | – | – | 15.3 | 23.2 | – | – | – | – | – | 7.8 | – |
| Bezymyannoye | 73 | – | – | 16 | – | – | – | – | – | – | 11 | – |
| Maloye Solovoye | 69 | – | – | 18 | – | – | 3 | – | – | – | 10 | – |
| Mokhovoye | 82 | 1.6 | – | 8 | – | – | – | – | – | – | 8.4 | – |
| Maloye Kharlamovo | 26.5 | 16.5 | 25 | 6.6 | – | 4.7 | 1 | 19.7 | – | – | – | – |
| Gagar'ye | 16.2 | 26.1 | 37.3 | 5.1 | 7.6 | 5.8 | – | – | – | – | 1.9 | – |

Note: Feldspars in total (albite, anorthite, anorthoclase).

**Table A5.** Results of Kruskal–Wallis one-way ANOVA by ranks and the median test for five groups of the semiarid lakes of the southwest of Western Siberia (N = 43).

| Parameter | Kruskal–Wallis Test | | Median Test | | |
|---|---|---|---|---|---|
| | **H** | ***p*** | **$\chi^2$** | **df** | ***p*** |
| pH | 3.75 | 0.441 | 2.33 | 4 | 0.675 |
| TDS | 17.0 | 0.002 * | 14.9 | 4 | 0.005 * |
| $Mg^{2+}$ | 16.9 | 0.002 * | 16.4 | 4 | 0.003 * |
| $Ca^{2+}$ | 19.5 | 0.0006 * | 14.7 | 4 | 0.005 * |
| $Na^+$ | 17.1 | 0.002 * | 16.9 | 4 | 0.002 * |
| $K^+$ | 20.4 | 0.0004 * | 18.8 | 4 | 0.0008 * |
| $Cl^-$ | 18.4 | 0.001 * | 14.9 | 4 | 0.005 * |
| $SO_4^{2-}$ | 22.4 | 0.0002 * | 16.4 | 4 | 0.003 * |
| DOC | 14.3 | 0.007 * | 16.4 | 4 | 0.003 * |
| DIC | 21.1 | 0.0003 * | 9.61 | 4 | 0.05 * |

* Significant at $p < 0.05$.

**Table A6.** Spearman correlation coefficients between morphological parameters of lakes and physico-chemical properties of waters (N = 43).

| | Area | Watershed | Average Depth | pH | TDS | $Mg^{2+}$ | $Ca^{2+}$ | $Na^+$ | $K^+$ | $Cl^-$ | $SO_4^{2-}$ | DOC | DIC |
|---|---|---|---|---|---|---|---|---|---|---|---|---|---|
| Area | 1 | | | | | | | | | | | | |
| Watershed | 0.96 * | 1 | | | | | | | | | | | |
| Average Depth | −0.08 | −0.05 | 1 | | | | | | | | | | |
| pH | −0.06 | 0.09 | 0.11 | 1 | | | | | | | | | |
| TDS | 0.49 * | −0.07 | −0.26 | −0.14 | 1 | | | | | | | | |
| $Mg^{2+}$ | 0.52 * | 0.47 * | −0.25 | −0.29 | 0.93 * | 1 | | | | | | | |
| $Ca^{2+}$ | 0.41 * | 0.5 * | −0.22 | −0.53 * | 0.57 * | 0.76 * | 1 | | | | | | |
| $Na^+$ | 0.51 * | 0.37 * | −0.26 | −0.19 | 0.97 * | 0.96 * | 0.61 * | 1 | | | | | |
| $K^+$ | 0.46 * | 0.5 * | −0.14 | 0.05 | 0.82 * | 0.8 * | 0.44 * | 0.84 * | 1 | | | | |
| $Cl^-$ | 0.52 * | 0.48 * | −0.3 | −0.22 | 0.97 * | 0.95 * | 0.62 * | 0.98 * | 0.82 * | 1 | | | |
| $SO_4^{2-}$ | 0.49 * | 0.5 * | −0.33 * | −0.18 | 0.93 * | 0.93 * | 0.61 * | 0.94 * | 0.77 * | 0.93 * | 1 | | |
| DOC | 0.36 * | 0.48 * | −0.11 | −0.04 | 0.9 * | 0.81 * | 0.4 * | 0.89 * | 0.86 * | 0.86 * | 0.79 * | 1 | |
| DIC | 0.26 | 0.4 * | −0.01 | 0.07 | 0.65 * | 0.57 * | 0.15 | 0.67 * | 0.73 * | 0.64 * | 0.62 * | 0.75 * | 1 |

* Significant at $p < 0.05$.

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
