# Peer review of "Semiarid Lakes of Southwestern Siberia as Sentinels of On-Going Climate Change: Hydrochemistry, the Carbon Cycle, and Modern Carbonate Mineral Formation"

_atmosphere, doi:10.3390/atmos14111624_

Round 1
Reviewer 1 Report
Comments and Suggestions for Authors
The paper concerns a very important question of natural variability of carbon cycle features in representative ecosystem. The study of natural variability provides an understanding of the possibility for generilize the areas in terms of spatial generalizations. The semiarid environments with abundant closed lakes represent very interesting object for a study of authigenic minerals formation, which was benefitially used by the authors of the publication. Heterogeniety of the carbonate minerals and their formation conditions were adressed and examined in context of the carbon sequestration.

Author Response
The paper concerns a very important question of natural variability of carbon cycle features in representative ecosystem. The study of natural variability provides an understanding of the possibility for generalize the areas in terms of spatial generalizations. The semiarid environments with abundant closed lakes represent very interesting object for a study of authigenic minerals formation, which was beneficially used by the authors of the publication. Heterogeneity of the carbonate minerals and their formation conditions were addressed and examined in context of the carbon sequestration.
We thank the reviewer for positive evaluation of our work.
Comments retrieved from the pdf file and our responses:
L 171 The TOC-V system was adopted to high salinity samples, or the samples were diluted prior to analysis?
The samples were diluted by a factor of 3 to 10 prior to analysis; added accordingly.
Fig 2, L 245 It should be indicated somewhere that CO32-+ HCO3- are represented by DIC.As I understand the DIC values were not recalculated to bicarbonate and carbonate concentrations?
This is very pertinent remark. Indeed, we did not distinguish between carbonate and bicarbonate ions; added to the Figure caption.
L306 Perhaps, the extra bracket
We rearranged this part of the text for clarity
L336-346 This is rather a part of the Discussion
We generally agree with this comment; however we do not explicitly discuss the clay minerals and iron hydroxides formation in this work, which is devoted to hydrochemistry and carbonate minerals (in the relevance to the topic of this Special Issue). For this reason, we preferred to briefly comment these findings right after their presentation in the results, instead of developing more extensive discussion in the relevant section.
L449 by the variations
Fixed; thanks for catching this!
L478 Remove the extra brackets
Here we used common notation of concentration; revised for clarity
L497 at a Mg/Ca – Fixed
L517 cases – Fixed
L518 observed in studied water bodies – Corrected
L524 presents, corrected
L525 The certain way of biogenic origin of EPS films was not satisfactory explained in the text
This is very pertinent remark. The EPS-related Mg-rich carbonate minerals are evidenced in Fig 8 c, d as mentioned in the relevant caption. Following this remark, we developed necessary description/possibility in the revised text.
L533 where – corrected
L536 Abundance of Mg-rich carbonates; revised accordingly
L537 Comprehensive; revised as recommended
L552-553 Is the data on organic matter in sediments taken from literature? In this work only dissolved compounds were likely being analyzed. If so, the reference should be added
This is good point. The statement about organic matter in sediments is based on the results of our macroscopic examinations, as well as microscopic studies of bottom sediments across the discussed groups. The fact that the bottom sediments of lakes within the Ishim Plain are represented by predominantly organic-rich sapropels was previously recognized [19, 26]. We added this information in the end of section 4.1.
L555 content of DOC; corrected accordingly
L566 as well as; corrected accordingly
Reviewer 2 Report
Comments and Suggestions for Authors
This is an interested work. However, some minor revisions are needed. In "Water chemistry as a factor of mineral formation" section, I suggested author to do some regression analysis among Ca, Mg, DOC and DIC.

Author Response
This is an interested work. However, some minor revisions are needed.
We appreciate positive evaluation of our work and constructive comments of the reviewer
In "Water chemistry as a factor of mineral formation" section, I suggested author to do some regression analysis among Ca, Mg, DOC and DIC.
Following this suggestion, additional regression analysis was carried out between the most important components of carbon cycle in the lakes. The DOC was moderately correlated with Mg2+ (r2 = 0.59, p < 0.00001). However, there was no linear relationship of DOC with Ca2+ (r2 = 0.002, p = 0.785). The DIC was related to none of these inorganic components (r ² < 0.12, p > 0.02), which likely stems from complex salt composition and history of formation of studied lakes. We added this new information to the end of revised section 3.1.
Minor comments retrieved from the marked pdf file
L41-43 We revised this sentence as following: “The water resources of Central Eurasia, located in the south of Western Siberia and the Northern Kazakhstan, within a territory of more than 1,600,000 km² [1], are highly vulnerable in terms of their hydrological regime which is impacted by on-going climate warming.”
L143 regions; corrected.
Reviewer 3 Report
Comments and Suggestions for Authors
The article presents conditions relating generally to the hydrochemical conditions of the Southwestern Siberia group of lakes. The paper has a well-organized layout, with detailed descriptions of the methods of study and the stages of sampling and analysis. I recommend the study for publication, after taking into account the following comments:
There is a reference to climate change in the title. Unfortunately, I do not find information about the course of air temperature in the analyzed region. What is the trend of these changes? How does the year in which the field research was done (2018 and 2019) compare to a longer period?
What is the length of ice cover in each group of lakes? The research was conducted during the summer, but as is well known, the length of the ice season has an impact on the characteristics of the lakes in the other months as well.
Please describe the structure of use of the analyzed catchments. What effect does this have on the results obtained?
The conclusions chapter should include information about the limitations of the conducted research and also about the further prospect of conducting it. How do the authors see the applicability of the obtained results?
Author Response
The article presents conditions relating generally to the hydrochemical conditions of the Southwestern Siberia group of lakes. The paper has a well-organized layout, with detailed descriptions of the methods of study and the stages of sampling and analysis. I recommend the study for publication, after taking into account the following comments:
We thank the reviewer for kind words and we revised the paper following his/her remarks as detailed below.
There is a reference to climate change in the title. Unfortunately, I do not find information about the course of air temperature in the analyzed region. What is the trend of these changes? How does the year in which the field research was done (2018 and 2019) compare to a longer period?
The two sampled years were within the mean multi-annual air temperature and precipitation. Indeed, mean annual air temperatures of 2018 and 2019 years were 2.1 and 3.3 °C, respectively, while for the historic period of observations the MAAT is 2.9 °C. Mean annual precipitation values of 2018 and 2019 were 343 and 406 mm respectively, which is comparable with mean mean-multi-annual value of 391 mm. These values have to be considered with caution, given that the study area is rather large and extended; as such, the data obtained at the weather station of the largest regional center (Kurgan) cannot be directly extrapolated to remote parts of the studied territory. We added necessary information in the revised section 2.2.
What is the length of ice cover in each group of lakes? The research was conducted during the summer, but as is well known, the length of the ice season has an impact on the characteristics of the lakes in the other months as well.
This is very pertinent comment. The duration of the ice cover on studied lakes ranges from 150 to 180 days, depending on the mineralization, size, depth and exact localization within the study territory [26]. Added to revised section 2.1 (first paragraph).
Please describe the structure of use of the analyzed catchments. What effect does this have on the results obtained?
As it is stated in the text, the landscapes of the study territory are significantly transformed by anthropogenic activities, mainly, agriculture. The watersheds are covered with birch groves and plowed fields [30]. However, it was beyond the scope of the present study to quantify the impact of anthropogenic load (land use, % of agricultural area) on dissolved carbon and major ions in the lake waters. For this, more comprehensive study including major nutrients (N, P) in the water column and incoming streams would be necessary. We would expect here that different forms of dissolved nitrogen originated form fertilizers or sewage would not directly impact carbonate mineral formation, other than increasing the frequency of bloom of coastal biofilm growth. In contrast, excess phosphate leaching from the watershed may not only boost the algal activity, but directly inhibit CaCO3 precipitation in some places while affecting hydrous MgCO3 precipitation in a lesser degree.
This work is currently in progress by our group and a relevant publication will be disseminated.
The conclusions chapter should include information about the limitations of the conducted research and also about the further prospect of conducting it. How do the authors see the applicability of the obtained results?
The main limitation of the present study is its low seasonal resolution (lack of spring, autumn and winter measurements) when extensive exchange of CO2 with atmosphere could occur, as it is known from works on other regions in Siberia (i.e., Serikova et al., 2019). The obtained results, however, encompass the most active open water period and as such can be considered as representative for assessing atmospheric C sequestration potential in bottom sediments. Missing in these evaluations are organic C burial (i.e., Manasypov et al., 2022) which should be a focus if further research. We added necessary paragraph in the revised Conclusions.
Serikova, S.; Pokrovsky, O.S.; Laudon, H.; Krickov, I.V.; Lim, A.G.; Manasypov, R.M.; Karlsson, J. High carbon emissions from thermokarst lakes of Western Siberia. Nature Communications 2019, 10, Art No 1552. https://doi.org/10.1038/s41467-019-09592-1.
Manasypov, R.M.; Lim, A.G.; Aliev, V; Shevchenko, V.P.; Shirokova, L.S.; Karlsson, J.; Pokrovsky, O.S. Carbon and nitrogen storage in thermokarst lake sediments of WSL peatlands: impact of climate, permafrost and lake size. Biogeochemistry 2022, 159 (1), 69-86. DOI: 10.1007/s10533-022-00914-y.